# Modularity and determinants of a (bi-)polarization control system from free-living and obligate intracellular bacteria

**Matthieu Bergé[1], Sébastien Campagne[2], Johann Mignolet[1†], Seamus Holden[3‡], Laurence Théraulaz[1], Suliana Manley[3], Frédéric H-T Allain[2], Patrick H Viollier[1]\***

[1]Department Microbiology and Molecular Medicine, Institute of Genetics and Genomics in Geneva, Faculty of Medicine, University of Geneva, Geneva, Switzerland; [2]Institute of Molecular Biology and Biophysics, Eidgenössische Technische Hochschule Zürich, Zürich, Switzerland; [3]Laboratory of Experimental Biophysics, École Polytechnique Fédérale de Lausanne, Lausanne, Switzerland

**Abstract** Although free-living and obligate intracellular bacteria are both polarized it is unclear whether the underlying polarization mechanisms and effector proteins are conserved. Here we dissect at the cytological, functional and structural level a conserved polarization module from the free living α-proteobacterium *Caulobacter crescentus* and an orthologous system from an obligate intracellular (rickettsial) pathogen. The NMR solution structure of the zinc-finger (ZnR) domain from the bifunctional and bipolar ZitP pilus assembly/motility regulator revealed conserved interaction determinants for PopZ, a bipolar matrix protein that anchors the ParB centromere-binding protein and other regulatory factors at the poles. We show that ZitP regulates cytokinesis and the localization of ParB and PopZ, targeting PopZ independently of the previously known binding sites for its client proteins. Through heterologous localization assays with rickettsial ZitP and PopZ orthologs, we document the shared ancestries, activities and structural determinants of a (bi-) polarization system encoded in free-living and obligate intracellular α-proteobacteria.

**\*For correspondence:** patrick. viollier@unige.ch

**Present address:** [†]Biochemistry and Molecular Genetics of Bacteria (BGMB), Institutdes Sciences de la Vie, Université Catholique de Louvain, Louvain-la-Neuve, Belgium; [‡]Centre for Bacterial Cell Biology, Newcastle University, Newcastle, United Kingdom

**Competing interests:** The authors declare that no competing interests exist.

## Introduction

Polarity is an ancient trait underlying developmental patterning and morphogenesis in eukaryotes and bacteria (*Martin and Arkowitz, 2014*; *Shapiro et al., 2002*; *St Johnston and Ahringer, 2010*). Cell pole-organizing proteins arose more than once during bacterial evolution as indicated by the fact that distinct polarization mechanisms exist in different bacterial phyla (*Davis and Waldor, 2013*; *Kirkpatrick and Viollier, 2011*; *Strahl and Hamoen, 2012*; *Treuner-Lange and Søgaard-Andersen, 2014*). The Gram-negative α-proteobacterial lineage encompasses polarized free-living (*Davis and Waldor, 2013*; *Haglund et al., 2010*; *Hallez et al., 2004*; *Kirkpatrick and Viollier, 2011*) and obligate intracellular bacteria including the mitochondrial ancestors (*Andersson et al., 1998*; *Haglund et al., 2010*).

The fresh-water bacterium *Caulobacter crescentus* is a model system for the genetic analysis of α-proteobacterial cell polarity because polar differentiation is tightly coordinated with cell cycle progression and because of the availability of a myriad of genetic tools to study this species compared to the obligate intracellular (rickettsial) pathogens (*Figure 1A*)(*Curtis and Brun, 2010*; *Ely, 1991*). The *C. crescentus* predivisional cell features the flagellum and a pilus biosynthesis machine at the new pole and a stalk, a cylindrical extension of the cell envelope, at the old pole. Upon completion of cell division, the replicative stalked (ST) cell progeny begins chromosome replication and an asymmetric cell division cycle. By contrast, the motile and piliated swarmer (SW) cell progeny resides

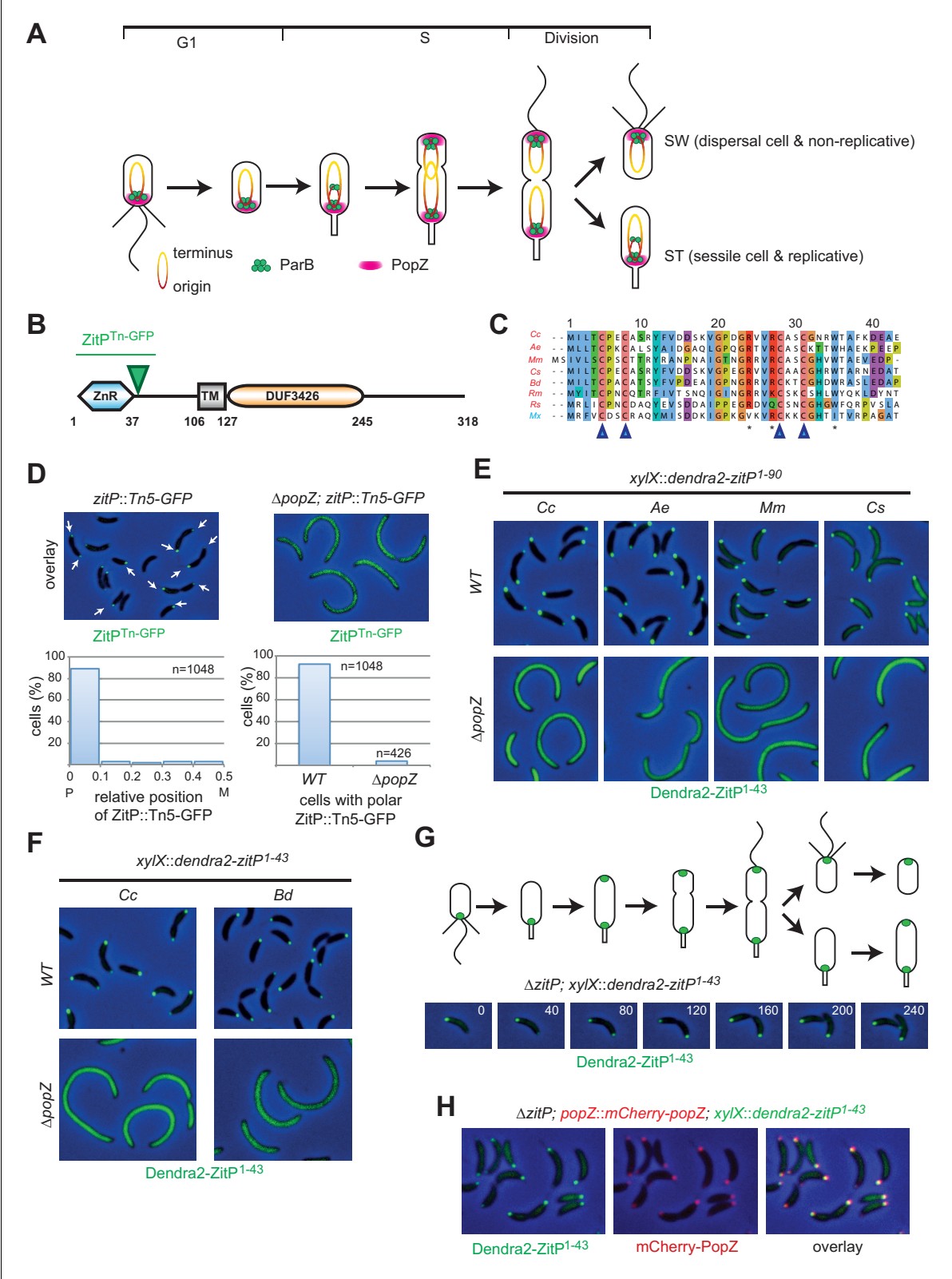

**Figure 1.** The Zinc finger (ZnR) of ZitP and orthologs is a polar localization signal. (A) Schematics of PopZ and ParB localization and chromosome organization during the *C. crescentus* cell cycle. Each cell cycle yields two different daughter cells: a swarmer (SW) and a stalked (ST) cell residing in G1- and S-phase, respectively. The replication origin region (red, including the centromeric sequence eight kbp from the origin) and the terminus region (yellow) are shown. (B) Schematic of the domain organization in ZitP: the N-terminal zinc-finger domain (ZnR), the transmembrane domain (TM) and the

*Figure 1 continued on next page*

Figure 1 continued

C-terminal domain-of-unknown function (DUF3426). The green arrowhead points to the codon in the *zitP* coding sequence harboring the GFP insertion in the *zitP::Tn5-GFP* strain. All regions are drawn to scale. Numbers indicate residues. (C) Alignment of the ZnR from α-proteobacterial ZitP orthologs (in red) and one δ-proteobacterium (in blue) (accession nos.: YP_002517671 [Cc, *Caulobacter crescentus*], ADU14901 [Ae, *Asticcacaulis excentricus*], ABI66665 [Mm, *Maricaulis maris*], ADG11315 [Cs, *Caulobacter segnis*], WP_003168465 [Bd, *Brevundimonas diminuta*], WP_014365322 [Rm, *Rickettsia massiliae*], WP_011909408 [Rs, *Rhodobacter sphaeroides*] and ABF87224 [Mx, *Myxococcus xanthus*]). The four cysteine residues coordinating the zinc ion are highlighted (blue arrowheads). Asterisks indicate the conserved residues promoting ZitP•PopZ complex formation. (D) Overlays of fluorescence and phase contrast images showing the subcellular localization of ZitP$^{Tn-GFP}$ encoded by the *zitP::Tn5-GFP* allele in *WT* or *ΔpopZ C. crescentus* cells (top). The graphs below show the quantitation of the localization from above. The left graph indicates the distribution of foci along the longitudinal axis. Focus (n = 1048) position is given in relative coordinates from 0 (pole) to 0.5 (midcell). P, pole; M, midcell. The right graph shows the percentage of cells containing at least one focus of ZitP$^{Tn-GFP}$ in *WT* (n = 1048) or in *ΔpopZ* cells (n = 426). (E) Overlay images as in D showing the subcellular localization of the first 90 residues of ZitP from *C. crescentus* (Cc) and orthologs from *A. excentricus* (Ae), *M. maris* (Mm), *C. segnis* (Cs) in *C. crescentus WT* (upper panels) or *ΔpopZ* (bottom panels) cells. Strains expressing Dendra2-ZitP$^{1-90}$ from the chromosomal *xylX* locus were induced with xylose for 4 hr before imaging (phase contrast and Dendra2-fluorescence). (F) Overlay images as in D showing the subcellular localization of the ZnR of ZitP (Dendra2-ZitP$^{1-43}$) of *C. crescentus* (Cc) and orthologs from *B. diminuta* (Bd) in *WT* (upper panels) or *ΔpopZ* (bottom panels) *C. crescentus* cells. Strains expressing Dendra2-ZitP$^{1-43}$ from the chromosomal *xylX* locus were induced with xylose for 4 hr before imaging (phase contrast and Dendra2-fluorescence). (G) Time-lapse imaging of swarmer cells from *ΔzitP* cells expressing Dendra2-ZitP$^{1-43}$ from the chromosomal *xylX* locus after induction for 1 hr with 0.3% xylose. Cells were then synchronized and transferred onto an agarose pad containing 0.3% xylose (t = 0 min), and visualized at 40 min intervals (time in minutes is indicated in the images) by phase contrast and Dendra2-fluorescence microscopy, respectively. Shown above the overlays are the schematics representing ZitP$^{1-43}$ localization during the *C. crescentus* cell cycle. (H) Images of *ΔzitP* cells expressing Dendra2-ZitP$^{1-43}$ from the chromosomal *xylX* locus and mCherry-PopZ from the native chromosomal *popZ* locus. Fluorescence and phase contrast images were acquired after 4 hr of induction with 0.3% xylose. Cells expressing Dendra2-ZitP$^{1-43}$ (left panel, green) and mCherry-PopZ (middle panel, red) are shown. Co-localized red and green foci appear yellow in the overlay (right panel).

The following figure supplement is available for figure 1:

**Figure supplement 1.** Conservation of ZitP and PopZ.

temporarily in a non-replicative (G1-like) state. At the SW to ST cell transition, the flagellated and piliated (SW) pole is remodeled into a ST pole and the developing cell acquires DNA replication competence. Replication of the circular chromosome proceeds bi-directionally from the single origin of replication (*Cori*) located at the nascent ST pole (*Curtis and Brun, 2010*). Once duplicated, the *Cori* region is rapidly segregated towards the nascent SW pole by the ParAB chromosome segregation system that targets the *parS* centromeric sequence located *circa* 8 kbp from *Cori* (*Figure 1A*) (*Mohl and Gober, 1997*; *Viollier et al., 2004*). The *cis*-encoded ParB protein binds *parS* and the resulting ParB•*parS* complex is guided pole-ward by the ParA ATPase, likely reinforced by poorly understood biophysical constraints and properties of the chromosome (*Lim et al., 2014*; *Mohl and Gober, 1997*). The PopZ polar organizing protein is thought to assemble a porous homo-polymeric matrix at the cell poles that captures the segregated ParB•*parS* complex (*Figure 1A*) via a direct interaction with ParAB (*Bowman et al., 2008*, *2013*; *Ebersbach et al., 2008*; *Holmes et al., 2016*; *Laloux and Jacobs-Wagner, 2013*).

The polar localization of PopZ is cell cycle-regulated: in newborn cells PopZ is localized to the old cell pole, whereas the newborn pole initially lacks a PopZ cluster (*Bowman et al., 2008*; *Ebersbach et al., 2008*). During S-phase, PopZ adopts a bipolar disposition (*Figure 1A*) coincident with ParB•*parS* segregation to facilitate its capture at the opposite pole. Formation of the second polar PopZ cluster may depend on the ParA ATPase and the TipN landmark protein, a coiled-coil protein that interacts with ParA and that marks the new pole as flagellar assembly site (*Huitema et al., 2006*; *Laloux and Jacobs-Wagner, 2013*; *Lam et al., 2006*; *Ptacin et al., 2010*). Although ParAB are essential for viability in *C. crescentus*, TipN or PopZ are individually dispensable (*Bowman et al., 2008*; *Ebersbach et al., 2008*; *Huitema et al., 2006*; *Lam et al., 2006*; *Mohl and Gober, 1997*), but joint inactivation of both genes arrests growth (*Schofield et al., 2010*). As most α-proteobacterial genomes encode PopZ and ParAB, but not TipN, TipN-independent control mechanism(s) for PopZ polarization should exist in α-proteobacteria.

In the Rhizobiales order of the α-proteobacteria, PopZ does not appear to fasten ParB at the cell poles as polar PopZ does not co-localize with ParB: while PopZ is monopolar, ParB localizes in a bipolar fashion (*Deghelt et al., 2014*; *Grangeon et al., 2015*). Thus, compared to the bipolar co-

localization of PopZ and ParB observed in *C. crescentus* (*Bowman et al., 2008*; *Ebersbach et al., 2008*; *Ptacin et al., 2014*), PopZ seems to have undergone functional specialization in the Rhizobiales, presumably interacting with other (unknown) client proteins. The genomes of the obligate intracellular (rickettsial) lineage also encode PopZ and ParAB orthologs (*Andersson et al., 1998*), but not several other known client proteins of *C. crescentus* PopZ that depend on a short N-terminal stretch in PopZ to interact with it (*Bowman et al., 2010*; *Holmes et al., 2016*; *Laloux and Jacobs-Wagner, 2013*; *Ptacin et al., 2014*).

Here, we unearth a reciprocal, physical and conserved interaction between PopZ and the cytoplasmic N-terminal zinc-finger domain (ZnR) from ZitP (*Hughes et al., 2010*), a bifunctional and bipolar membrane protein whose C-terminal DUF3426 domain is required for polar pilus biogenesis (*Mignolet et al., 2016*)(*Christen et al., 2016*), but dispensable for motility. We locate the structural determinants governing PopZ•ZitP complex formation and we show that this interaction is required to control cytokinesis and centromere positioning from the membrane. The PopZ•ZitP interaction differs for previously described interactions between PopZ and client proteins (*Holmes et al., 2016*) in that it does not require the aforementioned PopZ N-terminal segment. We show that ZitP induces PopZ bipolarity in the heterologous host *Escherichia coli* and we reconstitute a bipolar ZitP•PopZ•ParB tripartite complex in this system. Examining rickettsial PopZ and ZitP orthologs, we find that the PopZ•ZitP complex is modular and that the structure-function relationship is maintained in the obligate intracellular lineage.

## Results

### PopZ-dependent localization of the ZitP zinc-finger domain (ZnR) of α-proteobacteria

We previously isolated a *C. crescentus* strain bearing a Tn5-*GFP* insertion in the 5′-proximal third of the poorly characterized and conserved *CC_2215/CCNA_02298* gene (*Figure 1—figure supplement 1A*, dubbed *zitP*, zinc-finger targeting PopZ, *Figure 1B–D*). The resulting strain expresses a bipolarly localized and truncated ZitP-GFP fusion protein and is viable (*Hughes et al., 2010*). Consistent with this result, genome-wide transposon insertion sequencing (Tn-Seq) revealed that ZitP is not essential for viability, but required for pilus biogenesis and motility (*Christen et al., 2011*, *2016*; *Hughes et al., 2010*)(*Mignolet et al., 2016*). The predicted ZitP protein harbors an N-terminal zinc-finger domain (zinc_ribbon_5 or PF13719 superfamily, residues 1–37, henceforth ZnR, *Figure 1B–C*) and a trans-membrane segment (TM, residues 106–127) preceding the C-terminal DUF3426 (residues 128–245). While the DUF3426 is required for polar pilus biogenesis (*Mignolet et al., 2016*), it is not required for motility or to direct ZitP to the cell poles since in polar ZitP$^{Tn-GFP}$ the GFP moiety is fused in-frame to codon 49 of ZitP (green triangle in *Figure 1B*).

Quantitative live-cell fluorescence imaging of ZitP$^{Tn-GFP}$ in *zitP*::Tn5-*GFP* cells (n = 1048) revealed monopolar fluorescent foci in non-constricted cells, while bipolar foci are present in constricted cells (*Figure 1D*), suggesting that the ZnR and/or adjacent residues are necessary and sufficient for ZitP polarization and that bipolarization is cell cycle-regulated. We engineered strains expressing a Dendra2-ZitP$^{(1-90)}$ fusion protein (an N-terminal translational fusion of the Dendra2 fluorescent protein to only the cytoplasmic part of ZitP) from the xylose-inducible promoter (P$_{xyl}$) at the *xylX* locus in the wild-type (*WT*, NA1000) or Δ*zitP* background and found an identical localization pattern as for the ZitP$^{Tn-GFP}$ strain, indicating that residues 1–90 carry the necessary information for polar localization of ZitP (*Figure 1E*). To test if polarization is also a feature of ZitP from other α-proteobacteria, we expressed orthologous Dendra2-ZitP$^{(1-90)}$ variants in *WT* or Δ*zitP C. crescentus* cells and observed an identical localization pattern of these fusion proteins (*Figure 1E*). By contrast, the zinc-finger domain from the unrelated AgmX protein encoded in the δ-proteobacterium *Myxococcus xanthus* (*Nan et al., 2010*) is not polar in *C. crescentus* (*Figure 1—figure supplement 1B*), indicating that the zinc-finger domain of AgmX diverged.

Next, we asked if the highly conserved 43 residues encompassing the *C. crescentus* ZitP ZnR domain (*Figure 1C*, *Figure 1—figure supplement 1A*) suffice to direct Dendra2 to the cell poles. Expression of Dendra2-ZitP$^{(1-43)}$ from *xylX* in *C. crescentus WT* or Δ*zitP* cells indeed gave rise to cell cycle-dependent bipolar fluorescence (*Figure 1G*): while Dendra2-ZitP$^{(1-43)}$ is initially monopolar in G1-phase cells, it switches to a bipolar disposition during S-phase, 60 min after the release of G1

cells into fresh medium. Since this cell cycle localization pattern of Dendra2-ZitP$^{(1-43)}$ closely resembles that of the polar matrix protein PopZ (**Bowman et al., 2008**; **Ebersbach et al., 2008**), we co-expressed Dendra2-ZitP$^{(1-43)}$ and mCherry-PopZ in Δ*zitP* (*xylX::dendra-zitP*$^{1-43}$*popZ::mCherry-popZ*) cells and found that mCherry-PopZ co-localizes with Dendra2-ZitP$^{(1-43)}$ (**Figure 1H**). Importantly, ZitP$^{Tn-GFP}$, Dendra2-ZitP$^{(1-90)}$ and Dendra2-ZitP$^{(1-43)}$ are all delocalized (diffuse) in Δ*popZ* cells (**Figure 1D, E and F**) and immunoblotting revealed that the abundance of Dendra2-ZitP$^{(1-43)}$ is not significantly affected by the Δ*popZ* mutation (**Figure 1—figure supplement 1C**).

Knowing that polar localization of Dendra2-ZitP$^{(1-43)}$ depends on PopZ, we then tested if Dendra2-ZitP$^{(1-43)}$ is sequestered into the large monopolar PopZ patch ('plug') that forms when PopZ is overexpressed in *C. crescentus* (**Bowman et al., 2008**; **Ebersbach et al., 2008**). This PopZ 'plug' seems to exclude cytoplasmic macromolecular structures as ribosomes and genomic DNA from the cell poles, while capturing PopZ-interacting proteins (**Bowman et al., 2010**). We observed Dendra2-ZitP$^{(1-43)}$ and full length Dendra2-ZitP to adopt a monopolar 'plug'-like disposition when PopZ is overexpressed (**Figure 2A**, **Figure 2—figure supplement 1A**), suggesting that Dendra2-ZitP$^{(1-43)}$ is indeed recruited to PopZ 'plugs'.

To test whether PopZ is sufficient to recruit Dendra2-ZitP$^{(1-43)}$, we expressed both proteins in the γ-proteobacterium *E. coli* and observed that Dendra2-ZitP$^{(1-43)}$ indeed co-localizes with mCherry-PopZ (**Figure 2B**). By contrast, in the absence of PopZ, Dendra2-ZitP$^{(1-43)}$ remains diffuse (cytoplasmic, see below) in this heterologous host. All orthologous Dendra2-ZitP$^{(1-43)}$ variants tested share this pattern (with the exception of the *Rhodobacter sphaeroides* variant, see below), co-localizing with *C. crescentus* mCherry-PopZ in *E. coli* (**Figure 2B**). Thus, PopZ recruits the (predicted) ZitP ZnR and this property is conserved in most ZitP orthologs.

## NMR structure of ZitP ZnR and determinants for PopZ binding

To determine if *C. crescentus* ZitP$^{(1-43)}$ and PopZ interact directly, we separately purified soluble ZitP$^{(1-43)}$ and PopZ with a C-teminal-hexa-histidine (His$_6$)-tag from an *E. coli* overexpression system. In size exclusion chromatography (SEC), PopZ elutes with an apparent molecular mass of oligomers (~200 kDa), while ZitP$^{(1-43)}$ elutes as a monomer (**Figure 2—figure supplement 1B–D**). ZitP$^{(1-43)}$ and PopZ can be co-purified by SEC and the resulting ZitP$^{(1-43)}$•PopZ complex has a high apparent molecular mass, suggesting that ZitP binds oligomeric PopZ. Isothermal titration calorimetry (ITC) estimated the dissociation constant (K$_D$) between ZitP$^{(1-43)}$ and PopZ at 700 nM (**Figure 2C**), confirming the specific interaction between both proteins in vitro.

To identify determinants within ZitP that are required for the interaction with PopZ, we first resolved the NMR solution structure of ZitP$^{(1-43)}$ by following a classical nuclear Overhauser effect (NOE)-based approach (**Figure 2E**, **Supplementary file 1**, **Figure 2—figure supplement 1E**, see Materials and methods). The resonance assignment revealed that cysteine residues C5, C8, C28 and C31 are reduced and harbor typical Cβ chemical shifts of a zinc ion coordination module comprised between 31.58 ppm and 32.96 ppm (**Kornhaber et al., 2006**). The 20 NMR structures of ZitP$^{(1-43)}$ were overlaid with a root mean square deviation of 0.23 Å over the backbone atoms (**Supplementary file 1**), revealing two double-stranded antiparallel β-sheets (ββαββ), forming a 'crab claw' in which C5, C8, C28, C31 are located at the turns of β1/β2 and β3/β4 chelate a zinc ion (**Figure 2D–E**). This ZitP ZnR structure is unusual, though related to the family of the 'ribbon zinc-fingers' (zinc_ribbon_5 or PF13719 superfamily), albeit it lacks the additional β-strand typically located in between the two β-sheets.

Next, we monitored the interaction between $^{15}$N-labeled ZitP$^{(1-43)}$ and unlabeled PopZ to locate residues in ZitP$^{(1-43)}$ that are influenced by the interaction with PopZ (**Figure 2F**). This turned our attention to the aromatic side-chain of an invariant tryptophan at position 35 (W35, **Figure 2E**) that stacks above the zinc-coordination module of ZitP and that is replaced by a new species upon the addition of PopZ. W35 is interesting because the aromatic side chain at this position is replaced by an isoleucine in the primary structure of the AgmX zinc-finger (**Figure 1C**) that does not localize in *C. crescentus* (**Figure 1—figure supplement 1B**). Moreover, W35 is surrounded by a surface-exposed patch of basic residues (K18, R24, R27, R34, **Figure 2E**) and it is required to bind PopZ in vitro, as determined by ITC with a W35I mutant derivative of ZitP$^{(1-43)}$ [ZitP$^{(1-43)W35I}$, **Figure 2C**]. Consistent with these ITC experiments, W35I impairs the interaction of Dendra2-ZitP$^{(1-43)}$ with polar PopZ in vivo (**Figure 3A and B**), as indicated by the diffuse fluorescence of Dendra2-ZitP$^{(1-43)}$ in *C. crescentus* and in *E. coli* cells expressing PopZ (**Figure 3A–B**). Similarly, a C5S/C8S/C28S/C31S

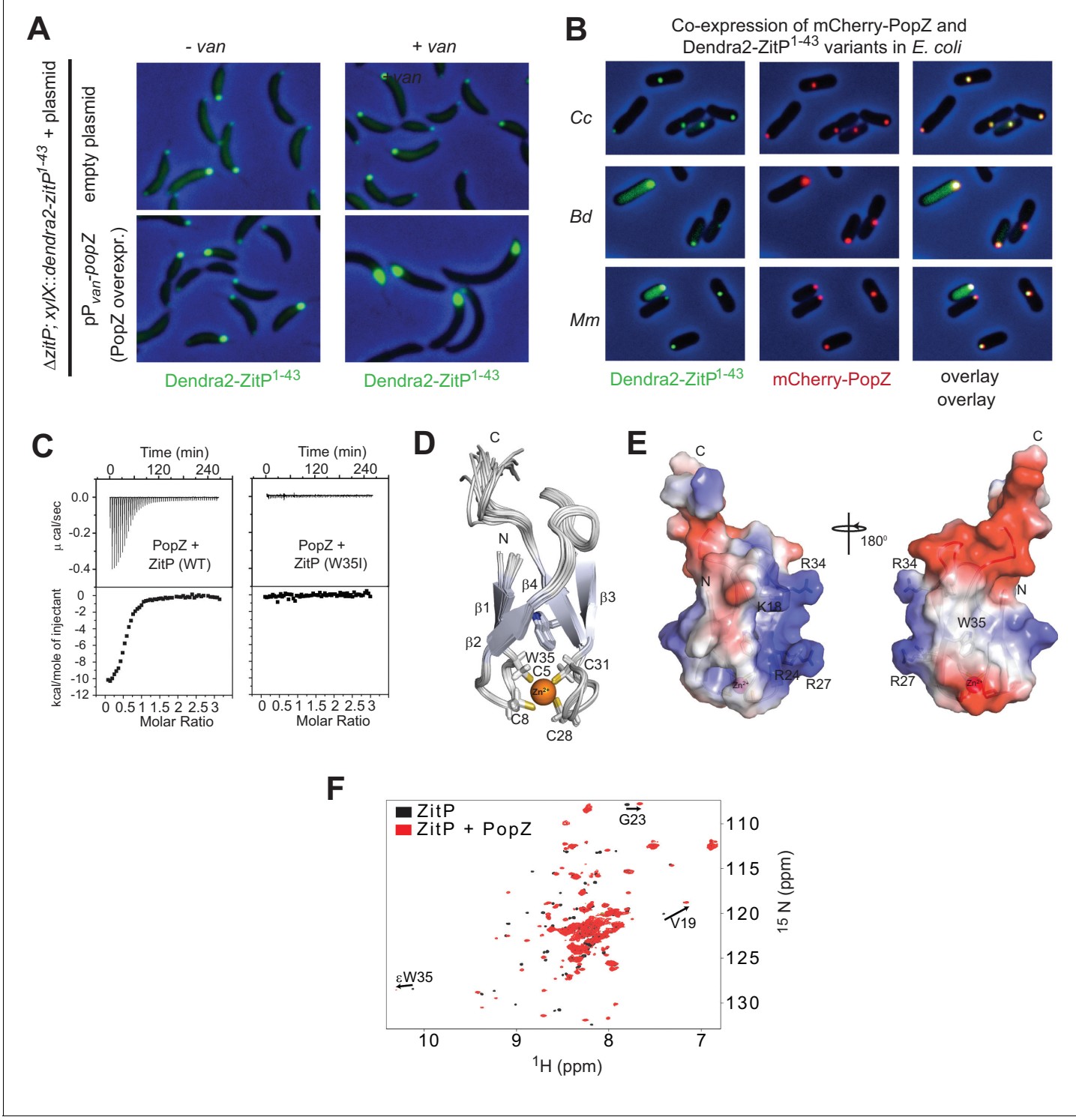

**Figure 2.** The ZnR of ZitP directly interacts with PopZ. (**A**) Images showing the localization of Dendra2-ZitP[1-43] in Δ*zitP* cells harbouring the empty vector (upper panel) or a plasmid to overproduce PopZ under control of the vanillate-inducible P_*van* promoter (lower panel). Images were taken before (- van) or after PopZ overexpression was induced by the addition of 0.5 mM vanillate for 5 hr (+ van). (**B**) Images of *E. coli* TB28 cells co-expressing Dendra2-ZitP[1-43] from *C. crescentus* (Cc), *B. diminuta* (Bd) or *M. maris* (Mm) and mCherry-PopZ. The Dendra2-fluorescence (green channel, right), the mCherry-fluorescence (red, middle) or the combined fluorescence (yellow) channels are shown as overlays with phase contrast images. Cells were grown in LB media for 2 hr, then Dendra2-ZitP[1-43] variants were induced with 1 mM IPTG and mCherry-PopZ was induced with 0.2% L-arabinose for 2 hr. (**C**) Isothermal titration calorimetry experiments (upper) measuring changes upon injection of 4 μL of a 300 μM PopZ solution into a 15 μM solution

*Figure 2 continued on next page*

Figure 2 continued

of ZitP$^{1-43}$ (left panel) or ZitP$^{1-43W35I}$ (right panel). (Lower) Plot showing the integrated heat changes following each injection as a function of the molar ratio of PopZ to ZitP$^{1-43}$ (left panel) or ZitP$^{1-43W35I}$ (right panel). (D) Stereo view of the NMR solution structure of ZitP$^{1-43}$. The secondary structure elements, the cysteine residues coordinating the zinc ion and W35 are indicated. (E) Electrostatic surface potential representation of ZitP$^{1-43}$. Several residues in the basic patch are labelled. (F) Overlay of the 2D $^1$H-$^{15}$N TROSY HSQC spectra of ZitP$^{1-43}$ (black spectrum) or ZitP$^{1-43}$ in complex with PopZ (red spectrum). Black arrows indicate spectroscopic shifts/appearance of new species.

The following figure supplement is available for figure 2:

**Figure supplement 1.** Purification of ZitP and PopZ.

quadruple or a R24A/R27A double mutation both disrupt polarization of Dendra2-ZitP$^{(1-43)}$ in *C. crescentus* and in PopZ-expressing *E. coli* cells (*Figure 3A–B*). As all of the substituted Dendra2-ZitP$^{(1-43)}$ variants are stably expressed (*Figure 3—figure supplement 1A*), we conclude that these residues are required for the formation of a polar ZitP$^{(1-43)}$•PopZ complex.

Interestingly, *R. sphaeroides* Dendra2-ZitP$^{(1-43)}$ is also largely diffuse in *C. crescentus* (*Figure 3C*) or in PopZ-expressing *E. coli*, but some faint polar signals are noticeable as well (*Figure 3D*). Questioning the basis for this poor polar localization in *C. crescentus*, we noted that this ZitP ortholog contains a naturally-occurring substitution (corresponding to R27Q in *Figure 1C*) in the aforementioned basic patch. We therefore asked if polar localization could be improved if this substitution is 'corrected' (i.e. reversed) by a Q27R substitution of *R. sphaeroides* Dendra2-ZitP$^{(1-43)}$. This 'corrected' *R. sphaeroides* ZitP$^{(1-43)Q27R}$ version is indeed robustly polar in *C. crescentus* (*Figure 3C*) and in PopZ-expressing *E. coli* cells (*Figure 3D*). When the W35I substitution is introduced into this 'corrected' *R. sphaeroides* variant, polar localization is again lost without compromising abundance (*Figure 3—figure supplement 1B*). We conclude that W35, the basic patch and the integrity of the Zn$^{2+}$ coordinating center are required for the formation of a polar ZitP$^{(1-43)}$•PopZ complex.

## ZitP controls PopZ localization

As PopZ interacts with the centromere (*parS*)-binding protein ParB and is required to anchor *parS* at both *C. crescentus* poles (*Bowman et al., 2008*; *Ebersbach et al., 2008*; *Laloux and Jacobs-Wagner, 2013*; *Ptacin et al., 2014*), we tested if the ZitP•PopZ complex associates, directly or indirectly, with *parS in vivo*. To this end, we conducted chromatin immunoprecipitation-deep-sequencing (ChIP-Seq) using antibodies recognizing ParB (anti-ParB) or ZitP (anti-ZitP specific for either the N-terminus or the C-terminus, *Figure 4—figure supplement 1*). These experiments confirmed that ParB exclusively occupies the *parS* site in vivo and additionally showed that i) the ZitP•PopZ complex associates with two chromosomal regions flanking, but not overlapping, *parS* (by 2–3 kpb) and that ii) the ParAB interaction sites in PopZ are critical for this association (*Figure 4—figure supplement 1*).

As ZitP is perfectly positioned to influence the subcellular position of ParB and/or PopZ, we examined the localization of both proteins in strains lacking or overproducing ZitP. While inactivation of ZitP has a mild effect on CFP-ParB localization in *WT* cells (*Figure 4—figure supplement 2*), we observed aberrant CFP-ParB localization and perturbed cytokinesis in Δ*zitP* cells expressing variants of PopZ impaired in binding ParB and/or ParA (*Laloux and Jacobs-Wagner, 2013*; *Ptacin et al., 2014*)(*Figure 4A*). Quantification revealed that Δ*zitP* Δ*popZ* cells expressing mCherry-PopZ$^{KE}$ (impaired in ParB binding, Δ*zitP mCherry-popZ$^{KE}$*) had fewer bipolar foci compared to *zitP$^+$* cells and an increase of cells without polar signals. Importantly, these abnormalities were not seen in *zitP$^+$* cells, indicating that ZitP promotes polar localization of PopZ when the ParAB-dependent (and/or another) localization pathway requiring the PopZ N-terminal region is impaired. Consistent with these results, we found that the efficiency of plating (EOP) of Δ*zitP* cells harboring the *popZ$^{Δ26}$* allele (Δ*zitP mCherry-popZ$^{Δ26}$*) that encodes a PopZ variant lacking the N-terminal 26 residues to bind ParAB and other client proteins (*Holmes et al., 2016*; *Ptacin et al., 2014*) is markedly reduced compared to *zitP$^+$* cells (*Figure 4B*). EOP was similarly diminished in Δ*zitP* Δ*popZ* cells expressing PopZ-$^{KEP}$ (Δ*zitP mCherry-popZ$^{KEP}$*) a version of PopZ in which the interaction sites for ParAB (and other PopZ client proteins) are inactivated by site-directed muatgenesis [E12K/R19E/S22P](*Figure 4B*).

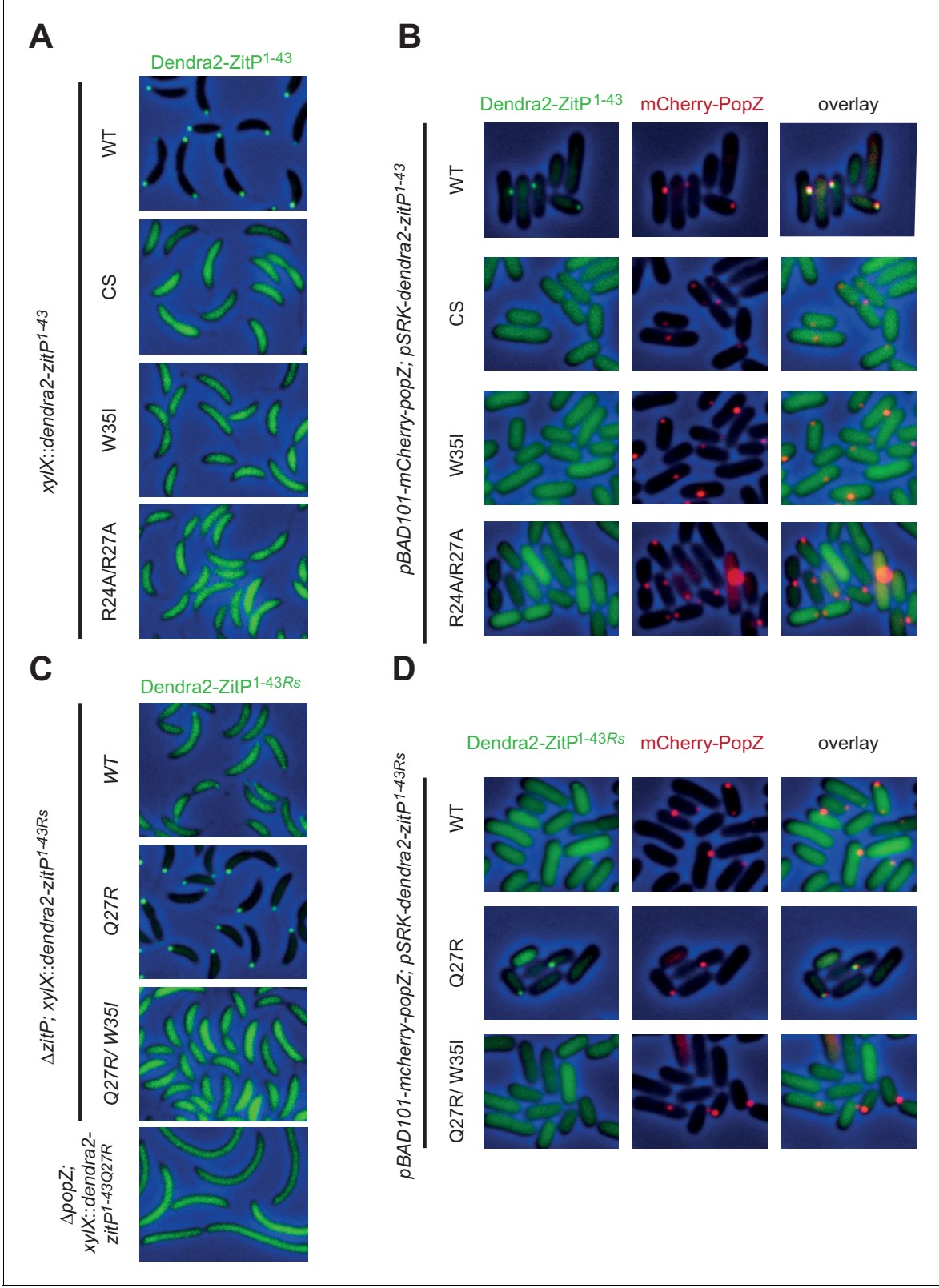

**Figure 3.** Molecular determinants underpinning the ZitP•PopZ complex. (**A**) Images showing the subcellular localization of *WT C. crescentus* cells expressing Dendra2-ZitP[1-43(WT)], Dendra2-ZitP[1-43(CS)], Dendra2-ZitP[1-43(W35I)] or Dendra2-ZitP[1-43(R24A/R27A)] from the *xylX* locus. Synthesis of the Dendra2-ZitP[1-43] variants was induced with xylose for 4 hr before phase contrast and Dendra2-fluorescence imaging. (**B**) Images of *E. coli* TB28 cells expressing Dendra2-ZitP[1-43(WT)], Dendra2-ZitP[1-43(CS)], Dendra2-ZitP[1-43(W35I)] or Dendra2-ZitP[1-43(R24A/R27A)] (left panels) in the presence of mCherry-PopZ (middle

Figure 3 continued

panels). Overlays between green (Dendra2) and/or red (mCherry) fluorescence and phase contrast images are shown (right panels). Cells were grown in LB for 2 hr, then the expression of Dendra2-ZitP$^{1-43}$ variants and of mCherry-PopZ was induced with 1 mM IPTG and 0.2% L-arabinose, respectively, for 2 hr. (C) Images of *WT C. crescentus* cells expressing *R. sphaeroides* (Rs) Dendra2-ZitP$^{1-43(WT)}$, Dendra2-ZitP$^{1-43(Q27R)}$ or Dendra2-ZitP$^{1-43(Q27R/W35I)}$ and of Δ*popZ C. crescentus* cells expressing *R. sphaeroides* Dendra2-ZitP$^{1-43(WT)}$. Synthesis of the Dendra2-ZitP$^{1-43}$ variants was induced from the *xylX* locus 4 hr before phase contrast and Dendra2-fluorescence imaging. (D) Images of *E. coli* TB28 cells expressing *R. sphaeroides* (Rs) Dendra2-ZitP$^{1-43(WT)}$, Dendra2-ZitP$^{1-43(Q27R)}$ or Dendra2-ZitP$^{1-43(Q27R/W35I)}$ (left panels) in the presence of mCherry-PopZ (middle panels). Overlays between green (Dendra2) and/or red (mCherry) fluorescence channels with phase contrast images are shown (right panel). Cells were grown in LB for 2 hr, then the expression of Dendra2-ZitP$^{1-43}$ variants and of mCherry-PopZ was induced with 1 mM IPTG and 0.2% L-arabinose, respectively, for 2 hr.

The following figure supplement is available for figure 3:

**Figure supplement 1.** Steady-state levels of Dendra-ZitP$^{1-143}$ variants.

While exploring which region in ZitP is required to control PopZ, we noted that Δ*zitP* Δ*popZ* cells expressing the mCherry-PopZ$^{KE}$ have a reduced growth rate in broth and that this defect is rescued by expression of (full-length) Dendra2-ZitP from the *xylX* locus using the xylose-inducible P$_{xyl}$ promoter (*Figure 4C*). The C-terminal DUF3426 that is required for polar pilus biogenesis (*Mignolet et al., 2016*) is not required to improve growth , as revealed by growth measurements of cells expressing the ZitP$^{(1-133)}$ variant. However, when the interaction with PopZ is abrogated [in the ZitP$^{(1-133W35I)}$ mutant], cells grow poorly. By contrast, no dependency on ZitP is seen in Δ*popZ* cells expressing otherwise unmodified mCherry-PopZ (*mCherry-popZ*, *Figure 4C*). Taken together, these results show that ZitP is important for growth and viability for Δ*popZ* cells expressing PopZ that no longer interacts efficiently with ParAB, and that the ability of membrane-anchored ZitP to bind PopZ via the ZnR is important to regulate PopZ. Thus, there exist at least two (redundant) mechanisms of PopZ localization control in *C. crescentus*: one regulated directly or indirectly by ParAB (*Laloux and Jacobs-Wagner, 2013*; *Ptacin et al., 2014*) (or the N-terminal region of PopZ) and another that is modulated by membrane-anchored ZitP [ZitP$^{(1-133)}$].

Further insight for the role of this truncated (membrane-anchored) ZitP came from overexpression of ZitP$^{(1-133)}$ from P$_{xyl}$ on a multi-copy plasmid (pMT464) in *cfp-parB mCherry-popZ* cells. We found that localization of ParB and PopZ is strongly perturbed (*Figure 4D*) upon induction of ZitP$^{(1-133)}$ with xylose. Time lapse-fluorescence imaging revealed that the pole-ward movement of CFP-ParB normally seen in S-phase of *WT* cells stalls halfway through in cells overexpressing ZitP$^{(1-133)}$ (*Figure 4D*, see *Figure 4—figure supplement 3A* for GFP-ParB). In these cells, mCherry-PopZ forms internal clusters, often proximal to CFP-ParB clusters. By contrast, CFP-ParB movement and mCherry-PopZ localization proceeds normally through the cell cycle in cells overexpressing the W35I derivative of ZitP$^{(1-133)}$ (*Figure 4D*). Also, ZitP$^{(1-133)}$ overexpression displaces a monomeric ParA derivative [ParA(G16V)] whose localization is PopZ-dependent (*Ptacin et al., 2014*) from the cell poles into aberrant cytoplasmic clusters (*Figure 4—figure supplement 3B*). These defects seen at the single cell level are accompanied by cell filamentation at the population level and are not apparent either with full-length ZitP (*Figure 4E*) overexpressed to comparable levels (*Figure 4—figure supplement 3C*) or with ZitP$^{(1-133)}$ variants harboring the W35I single mutation, the R24A/R27A double mutation or the C5S/C8S/C28S/C31S quadruple mutation (*Figure 4E*). As overexpression of the ZnR without any membrane anchoring domain [ZitP$^{(1-90)}$] does not cause such cell cycle defects either (*Figure 4F*), we asked if membrane-anchoring is specific for this ZitP activity. The aforementioned overexpression phenotypes are still supported by overexpression of a ZitP version in which the TM of ZitP$^{(1-133)}$ had been replaced with an unrelated TM from the *E. coli* MalF protein (*Figure 4F*). Finally, we also observed that overproduction of ZitP$^{1-133}$ in *mCherry-popZ$^{KEP}$* cells [in which the interaction with ParAB is impaired, (*Laloux and Jacobs-Wagner, 2013*; *Ptacin et al., 2014*) recapitulates the overexpression phenotype (*Figure 4G*) seen in *mCherry-popZ* cells. Taken together, these results show that ZitP is able to redirect PopZ localization via a direct interaction at the membrane independently of the known interaction sites for ParA and ParB (and/or potentially other PopZ client proteins). As a consequence of PopZ delocalization, the localization of DNA segregation proteins like ParB and ParA is perturbed and cells divide infrequently owing to the tight coupling of chromosome segregation and cytokinetic control (*Thanbichler and Shapiro, 2006*). Since

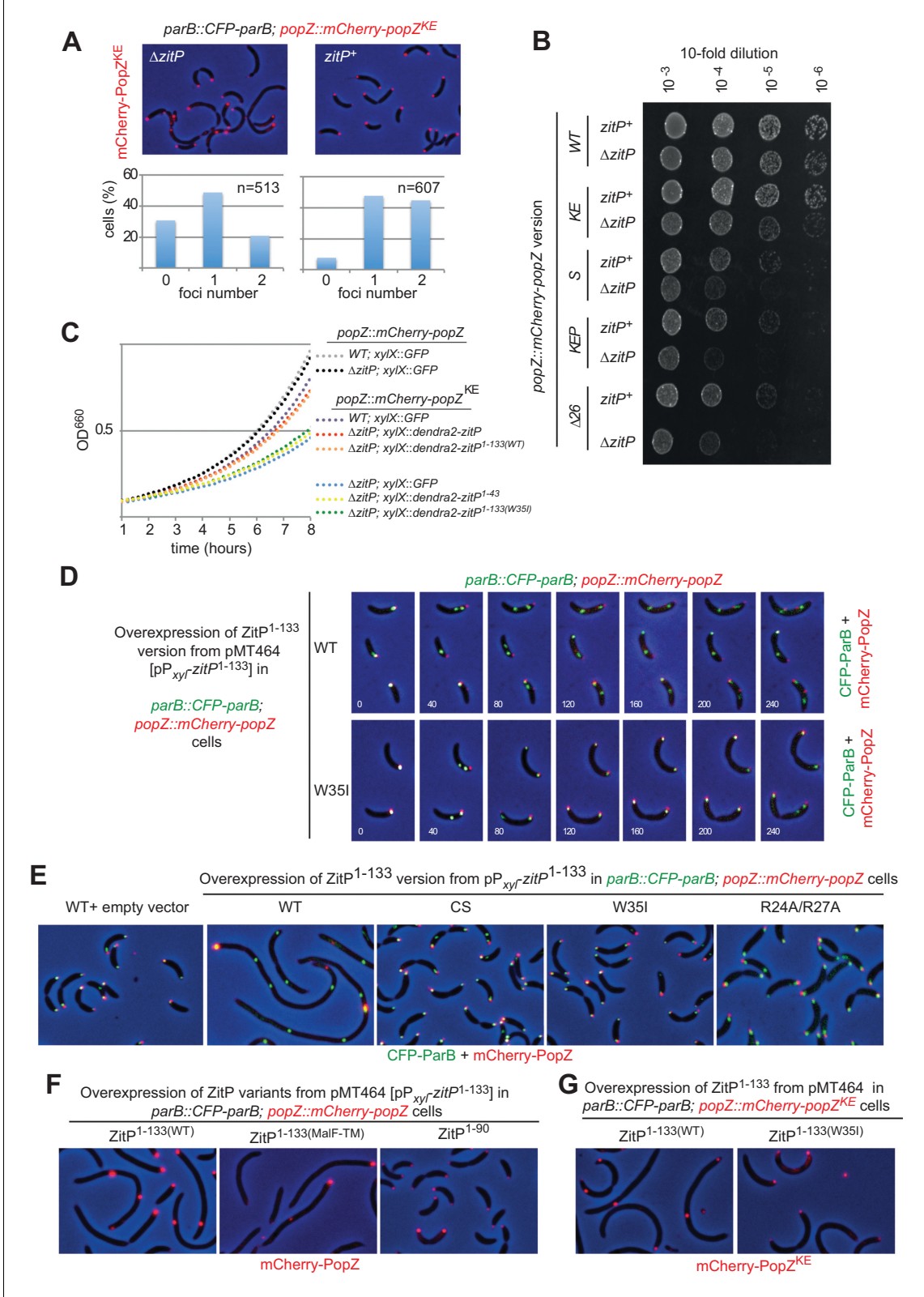

**Figure 4.** The ZitP•PopZ complex controls the *C. crescentus* cell division cycle. (**A**) Overlays of mCherry-fluorescence and phase contrast images of *WT* (upper panels) or Δ*zitP* (bottom) *C. crescentus* expressing the mCherry-PopZ$^{KE}$ variant that no longer interacts with ParB. mCherry-PopZ$^{KE}$ is expressed from the native locus (*mCherry-popZ$^{KE}$*) in lieu of untagged PopZ. Below the micrographs are quantifications of cells with and without bipolar or monopolar fluorescent foci of mCherry-PopZ$^{KE}$. (**B**) Efficiency of plating (EOP) assays of *C. crescentus* strains expressing *WT* mCherry-PopZ (*mCherry-*

*Figure 4 continued on next page*

Figure 4 continued

*popZ*) or variants that no longer interact with ParB (*mCherry-popZ^{KE}*), with ParA (*mCherry-popZ^{SP}*) or both (*mCherry-popZ^{KEP}* and *mCherry-popZ^{Δ26}*) in WT or *ΔzitP* cells. Serial ten-fold dilutions were plated on PYE agar containing spectinomycin. (C) Growth measurements of various strains monitored by optical density at 660 nm (OD660) in PYE. (D) Overlays of CFP- and mCherry-fluorescence with phase contrast images from *C. crescentus popZ:: mCherry-popZ parB::CFP-parB* cells harbouring an empty plasmid (pMT464, left panel) or the pP_{xyl}-ZitP^{1-133(WT)} derivative followed by time-lapse analysis with images acquired every 40 min. (E) Overlays of mCherry-fluorescence with phase contrast images from *C. crescentus popZ::mCherry-popZ parB::CFP-parB* cells harbouring an empty plasmid (pMT464, left panel) or derivatives: pP_{xyl}-ZitP^{1-133(WT)} (second panel), pP_{xyl}-ZitP^{1-133(CS)} (third panel), pP_{xyl}-ZitP^{1-133(W35I)} (fourth panel) and pP_{xyl}-ZitP^{1-133(R24A/R27A)} (right panel). Overexpression of ZitP^{1-133} variants was induced with xylose 0.3% for 6 hr prior to imaging. (F) Overlays of mCherry-fluorescence with phase contrast images from *C. crescentus popZ::mCherry-popZ parB::CFP-parB* cells harbouring a P_{xyl}-ZitP^{1-133(MalF-TM)}, P_{xyl}-ZitP^{1-133(WT)} or a P_{xyl}-ZitP^{1-90} overexpression plasmid. Overexpression was induced by growth in 0.3% xylose for 6 hr prior to imaging. (G) Overlays of mCherry-fluorescence with phase contrast images from *C. crescentus popZ::mCherry-popZ^{KEP} parB::CFP-parB* cells harbouring a P_{xyl}-ZitP^{1-133(WT)} or a P_{xyl}-ZitP^{1-133(W35I)} overexpression plasmid. Overexpression was induced by growth in 0.3% xylose for 6 hr prior to imaging.

The following figure supplements are available for figure 4:

**Figure supplement 1.** ChIP-Seq analysis of ZitP.

**Figure supplement 2.** Quantification of polar CFP-ParB and mCherry-PopZ in *C.crescentus WT* and mutants.

**Figure supplement 3.** Localization of ParB and ParA upon ZitP^{1-133} overexpression in *C.crescentus*.

*mCherry-popZ^{KEP}* cells are also affected by ZitP^{1-133} in growth and division, an unknown PopZ client protein(s) may contribute to the PopZ-dependent phenotypes caused by under- or over-expression of ZitP^{1-133}.

## ZitP imparts bipolarity upon PopZ in E. coli

Knowing that membrane-anchored ZitP controls PopZ localization in *C. crescentus*, we attempted to recapitulate these effects in a heterologous system by co-expression of ZitP^{(1-43)} or ZitP^{(1-133)} with mCherry-PopZ in *E. coli*. When ZitP^{(1-43)} is co-expressed with mCherry-PopZ in this system, mCherry-PopZ forms a single cluster (*Figure 5—figure supplement 1A*) resembling the ones previously described (*Bowman et al., 2008*; *Ebersbach et al., 2008*; *Laloux and Jacobs-Wagner, 2013*) whose location is fairly sporadic and infrequently associated with the cell extremity (*Figure 5—figure supplement 1A*). By contrast, co-expression of full-length ZitP or ZitP^{(1-133)} directs mCherry-PopZ into a robust bipolar disposition in *E. coli* cells (*Figure 5A*, *Figure 5—figure supplement 1A*). Conversely, the localization of Dendra2-ZitP in *E. coli* co-expressing PopZ provided matching results: Dendra2-ZitP^{(1-133)} and full length Dendra2-ZitP are both bipolar, whereas Dendra2-ZitP^{(1-133)W35I} is dispersed (*Figure 5B*). To confirm that Dendra2-ZitP associates with the polar membrane in *E. coli* rather than inclusion bodies, we localized it by super-resolution photo-activated localization microscopy (PALM) (*Betzig et al., 2006*) and observed the fluorescent signals to line exclusively the polar caps (*Figure 5C*). By contrast, Dendra2-ZitP is dispersed along the cell envelope when PopZ is absent (*Figure 5C*). Thus, these interdependent clusters of PopZ and ZitP at the polar caps are clearly distinct from the internal (cytoplasmic) aggregates typically seen for PopZ when expressed without ZitP in *E. coli* cells (*Figure 5A*) or in division-inhibited (filamentous) *E. coli* (*Laloux and Jacobs-Wagner, 2013*) and that resemble fluorescent aggregates of inclusion bodies (*Miller et al., 2015*).

Dendra2-ZitP^{(1-133)} also induces the redistribution of the ParAB-blind version of mCherry-PopZ (mCherry-PopZ^{Δ26}) in *E. coli* (*Figure 5—figure supplement 1B*). By contrast, Dendra2-ZitP^{(1-43)} is unable to do so, yet it still co-localizes (interacts) with mCherry-PopZ^{Δ26}. This indicates that Dendra2-ZitP^{(1-43)} can still interact with mCherry-PopZ^{Δ26}, but no longer controls its localization since Dendra2-ZitP^{(1-43)} is not membrane-anchored. Thus, ZitP regulates mCherry-PopZ localization either via the conserved C-terminal DUF2497 domain that is conserved in PopZ orthologs or via sequences distal to the N-terminal (26) residues where the ParAB recognition sites reside. As these findings imply that ZitP and ParB do not compete for the same binding site in PopZ, we reasoned that it should be possible to reconstitute a tripartite bipolar complex by co-expression of PopZ and Dendra2-ParB together with either mCherry-ZitP^{(1-133)} or mCherry-ZitP^{(1-133)W35I} in *E. coli* (*Figure 5D*).

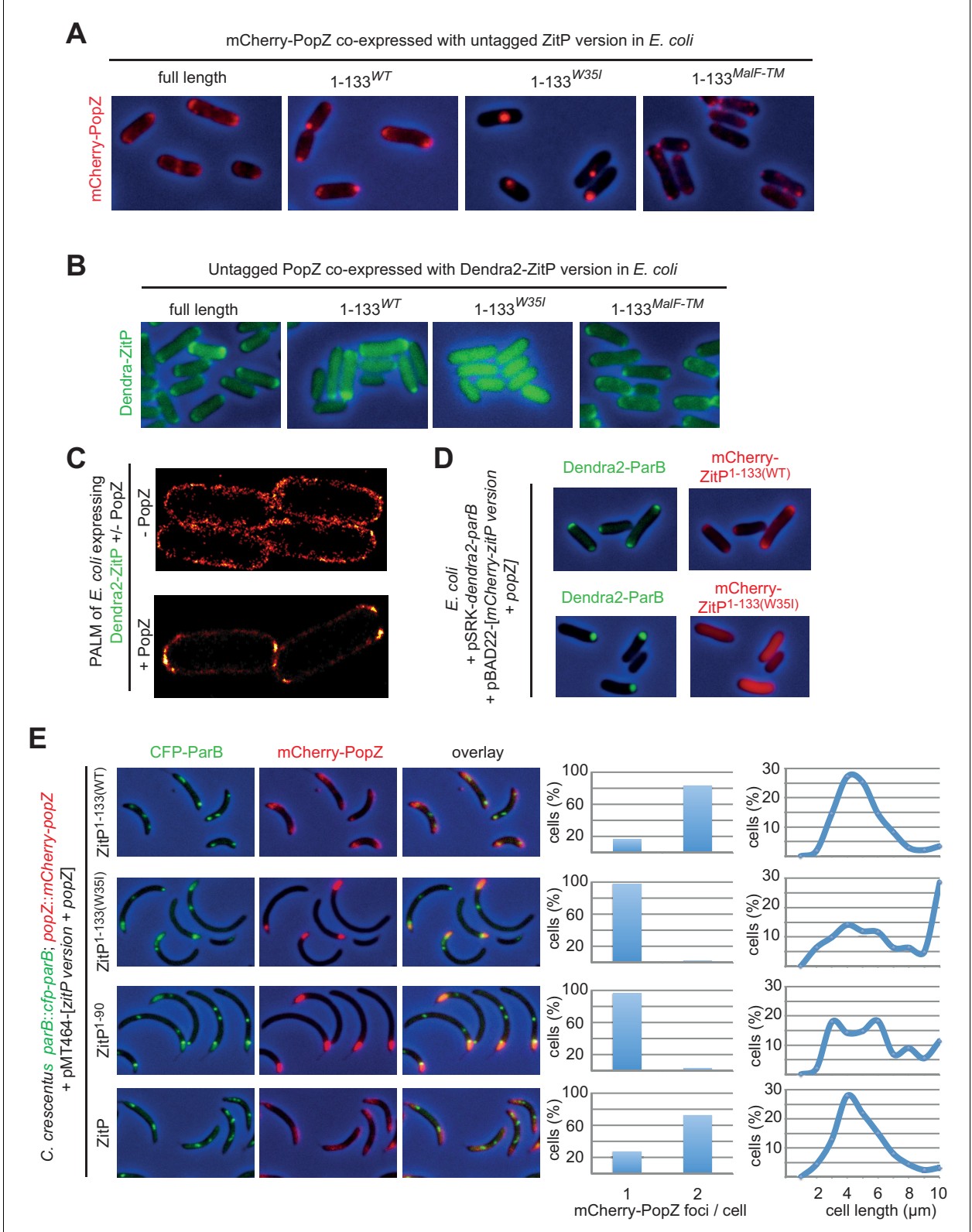

**Figure 5.** ZitP controls PopZ bipolarity in *E. coli* and *C. crescentus*. (**A**) Overlays of mCherry-fluorescence with phase contrast images of *E. coli* TB28 cells co-expressing mCherry-PopZ and various (untagged) ZitP versions: full length ZitP, or the derivatives ZitP$^{1-133}$, ZitP$^{1-133(W35I)}$ and ZitP$^{1-133(MalF-TM)}$. Cells were grown in LB for 2 hr, then ZitP variants and mCherry-PopZ were induced with 1 mM IPTG and 0.2% L-arabinose, respectively, for 2 hr before imaging. (**B**) Overlays of Dendra2-fluorescence with phase contrast images of *E. coli* TB28 cells co-expressing (untagged) PopZ and full length Dendra2-

*Figure 5 continued*

ZitP, Dendra2-ZitP$^{1-133(WT)}$, Dendra2-ZitP$^{1-133(W35I)}$ or Dendra2-ZitP$^{1-133(MalF-TM)}$. Cells were grown in LB for 2 hr, then Dendra2-ZitP and mCherry-PopZ were induced with 1 mM IPTG and 0.2% L-arabinose, respectively, for 2 hr before imaging. (C) PALM (photo-activated localization microscopy) images of *E. coli* cells expressing Dendra2-ZitP from pSRK-Km (*Khan et al., 2008*) and either no PopZ (empty pBAD101(*Guzman et al., 1995*) vector, strain EC127, left panel) or untagged PopZ from pBAD101 (strain EC132, right panel). (D) Overlays of Dendra2- and mCherry-fluorescence with phase contrast images showing the co-localization of Dendra2-ParB (from P$_{lac}$ on pSRK-Km) with WT or W35I mCherry-ZitP$^{1-133}$ derivatives in *E. coli* cells co-expressed with untagged PopZ from P$_{ara}$ on pBAD22. (E) Overlays of CFP- and/or mCherry-fluorescence with phase contrast images of *C. crescentus popZ:: mCherry-popZ parB::CFP-parB* cells co-overexpressing (untagged) full-length ZitP, ZitP$^{1-90}$, ZitP$^{1-133(W35I)}$ or ZitP$^{1-133(WT)}$ with (untagged) PopZ under P$_{xyl}$ control from pMT464. Over-expression was induced by growth in xylose 0.3% for 4 hr prior to imaging. mCherry-PopZ (upper panel) and CFP-ParB (middle panel) are expressed from their native promoters at the respective endogenous chromosomal loci in lieu of the untagged form.

The following figure supplements are available for figure 5:

**Figure supplement 1.** Co-localization of ZitP and PopZ in *E.coli*.

**Figure supplement 2.** Steady-state levels of PopZ and ZitP variants upon co-overexpression in *C.crescentus*.

We found that mCherry-ZitP$^{(1-133)}$ can indeed direct Dendra2-ParB to the poles of PopZ-expressing *E. coli*, whereas it remains in a cytoplasmic aggregate in the presence of mCherry-ZitP$^{(1-133)W35I}$.

These experiments in *E. coli* prompted us to examine if ZitP$^{(1-133)}$ can redistribute mCherry-PopZ from the monopolar 'plug' of overexpressed PopZ in *C. crescentus*. Fluorescence microscopy of *mCherry-popZ CFP-parB* cells harboring either a PopZ/ZitP or a PopZ/ZitP$^{(1-133)}$ co-overexpression plasmid (*Figure 5E*) revealed bipolar signals of mCherry-PopZ and CFP-ParB in the presence of ZitP$^{(1-133)}$. By contrast only monopolar 'plugs' containing mCherry-PopZ and CFP-ParB were seen with the W35I derived or ZitP$^{(1-90)}$ co-overexpression plasmids at similar expression levels (*Figure 5— figure supplement 2*). In the latter condition, there is co-localization between PopZ and ParB. However, ParB is no longer at the periphery of bipolar PopZ in cells harboring the PopZ/ZitP or the PopZ/ZitP$^{(1-133)}$ co-overexpression plasmid. Cell length quantification revealed that cells containing monopolar 'plugs' are longer (*Figure 5E*) than the PopZ/ZitP$^{(1-133)}$ co-overexpressing cells. As biochemical fractionation experiments confirmed the predicted localization of ZitP$^{(1-43)}$ and ZitP$^{(1-133)}$ (*Figure 5—figure supplement 1C*), we conclude that membrane-anchored ZitP robustly and directly controls PopZ localization in *E. coli* and in *C. crescentus*.

## ZitP control of PopZ is conserved

To investigate if the ZitP-dependent control of PopZ localization is common to closely and distantly related α-proteobacterial lineages, we overexpressed ZitP$^{(1-120)}$ from *Brevundimonas diminuta* and ZitP$^{(1-104)}$ from *Rickettsia massiliae* [corresponding to *C. crescentus* ZitP$^{(1-133)}$] in *C. crescentus* and found that both induced filamentation with mislocalized CFP-ParB and mCherry-PopZ akin to *C. crescentus* ZitP$^{(1-133)}$ (*Figure 6A*). As for *C. crescentus* ZitP or ZitP$^{(1-133)}$ (*Figure 5E*), ZitP$^{(1-104)Rm}$ prevents the accumulation of monopolar 'plugs' of overexpressed PopZ$^{Cc}$ in *C. crescentus* (*Figure 6B*). As expected, this *R. massiliae* ZnR [ZitP$^{(1-43)Rm}$] also localizes to the *C. crescentus* cell poles in a PopZ-dependent manner, but no longer when the analogous W35I mutation is introduced (*Figure 6C*). Rickettsial ZnR also co-localizes with *C. crescentus* mCherry-PopZ in *E. coli* (*Figure 6D*) and ZitP$^{(1-104)Rm}$ directs mCherry-PopZ into bipolarity (*Figure 6E*). Finally, we found that i) *R. massiliae* PopZ can also recruit *C. crescentus* Dendra2-ZitP$^{(1-43)}$ to clusters in *E. coli* (*Figure 6F*) and that ii) it promotes the bipolar localization of full-length Dendra2-ZitP (*Figure 6G*). We conclude that the PopZ-ZitP bipartite bipolarization system is conserved in α-proteobacteria and modular.

## Discussion

Eukaryotic zinc-finger (ZnR) domains are typically used to bind DNA, but they can also mediate protein-protein interactions (*Klug, 2010*). Here we described a small (43-residue) ZnR that acts as a conserved polar localization sequence, reminiscent of the nuclear localization signal (NLS) of eukaryotes, by promoting the interaction with the PopZ polar organizer of α-proteobacteria. This ZnR interacts

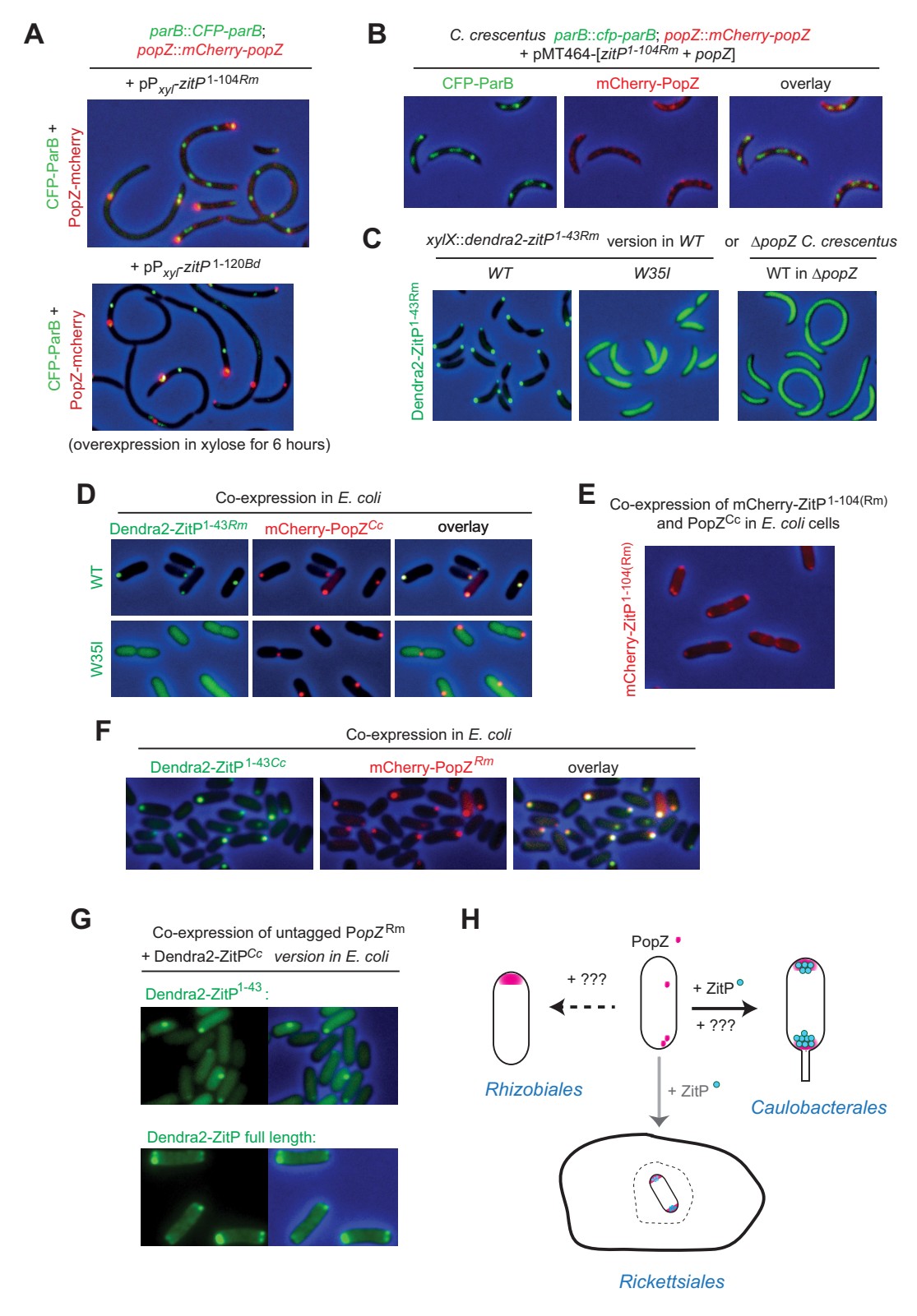

**Figure 6.** Conservation of PopZ•ZitP localization and activity (**A**) Overlays of CFP- and mCherry-fluorescence with phase contrast images of *C. crescentus popZ::mCherry-popZ parB::CFP-parB* cells over-expressing ZitP$^{1-104}$ from *R. massiliae* (upper panel) or *B. diminuta* (lower panel) from P$_{xyl}$ on a multi-copy plasmid (pMT464). mCherry-PopZ and CFP-ParB are expressed from the chromosome in lieu of the untagged versions. ZitP$^{1-104}$ over-expression was induced by the addition of 0.3% xylose for 6 hr prior to imaging. (**B**) Overlays of CFP- and mCherry-fluorescence with phase contrast

*Figure 6 continued on next page*

Figure 6 continued

images of *C. crescentus popZ::mCherry-popZ parB::CFP-parB* cells over-expressing ZitP$^{1-104}$ from *R. massiliae* with *C. crescentus* PopZ. Over-expression was induced by the addition of 0.3% xylose for 6 hr prior to imaging. (C) Overlays of Dendra2-fluorescence with phase contrast images of the *R. massiliae* (Rm) ZitP ZnR version expressed from the *xylX* locus in *WT* and Δ*popZ C. crescentus* cells. Synthesis of the Dendra2-ZitP$^{1-43(WT)Rm}$ or Dendra2-ZitP$^{1-43(W35I)\ Rm}$ was induced for 4 hr with 0.3% xylose before imaging. (D) Overlays of Dendra2- and/or mCherry-fluorescence with phase contrast images of *E. coli* TB28 cells expressing *R. massiliae* (Rm) Dendra2-ZitP$^{1-43(WT)}$ (upper panels), Dendra2-ZitP$^{1-43(W35I)}$ (lower panel) in the presence of mCherry-tagged PopZ from *C. crescentus*. Cells were grown in LB for 2 hr, then Dendra2-ZitP$^{1-43}$ variants and mCherry-PopZ were induced with 1 mM IPTG and 0.2% L-arabinose, respectively, for 2 hr. (E) Overlay of mCherry-fluorescence with phase contrast images of *E. coli* TB28 cells co-expressing mCherry-ZitP$^{1-104Rm}$ from *R. massiliae* (Rm) with *C. crescentus* PopZ from P$_{ara}$ encoded on the same pBAD22-derived plasmid. Cells were grown in LB for 2 hr, then expression of mCherry-ZitP$^{1-104Rm}$ and PopZ was induced with 0.2% L-arabinose for 2 hr before imaging. (F) Overlays of Dendra2- and/or mCherry-fluorescence with phase contrast images of *E. coli* TB28 cells expressing *C. crescentus* Dendra2-ZitP$^{1-43}$ (Cc) in the presence of mCherry-PopZ from *R. massiliae* (Rm). Cells were grown in LB for 2 hr, then Dendra2-ZitP$^{1-43}$ and mCherry-PopZ were induced with 1 mM IPTG and 0.2% L-arabinose, respectively, for 2 hr. (G) Images of *E. coli* cells co-expressing *C. crescentus* Dendra2-ZitP$^{1-43}$ (Cc, upper panel) or Dendra2-ZitP full-length (Cc, bottom panel) and PopZ from *R. massiliae* (Rm). Fluorescence (Dendra2) images (left panels) and overlays between phase contrast and Dendra2 fluorescence images (right panels) are shown. Cells were grown in LB to during 2 hr, then Dendra2-ZitP$^{1-43}$ variants were induced with 1 mM IPTG and PopZ was induced with 0.2% L-arabinose for 2 hr. (H) The (bi)polar PopZ•ZitP complex of free-living (Caulobacterales) and obligate intracellular (Rickettsiales) α-proteobacteria. Pink dots denote PopZ monomers that assemble into a bipolar or monopolar patch, while blue dots denote ZitP molecules. An obligate intracellular rickettsial (rod with bipolar PopZ) cell is depicted within a vacuole (dashed structure) of a host cell (closed structure) and presumed also to polarize PopZ•ZitP (grey arrow). As ZitP is not present in the Rhizobiales, another mechanism of PopZ control is likely operational to drive it into a monopolar disposition. Similarly, we suggest that PopZ localization in *C. crescentus* can be accomplished by another pathway that operates independently of ZitP and likely involves ParAB and/or another pathway (see Discussion).

directly with PopZ and controls its localization from the membrane, even when PopZ no longer carries the interaction sites for ParAB, indicating that PopZ localization is regulated by redundant mechanisms in *C. crescentus.* One mechanism may require ParAB or at least it depends on the N-terminal region of PopZ that promotes the interaction with ParAB and other PopZ client proteins (*Holmes et al., 2016*). However, ZitP controls PopZ independently of this region, underscoring the multivalent interactions and multifunctional properties of PopZ.

ZitP emerges as new a component of the α-proteobacterial polarity control pathway using an N-terminal ZnR to regulate PopZ localization. Interestingly, ZitP is bifunctional, controlling the localization of polar pilus biogenesis proteins via its C-terminal DUF3426 (*Mignolet et al., 2016*). While the DUF3426 is less conserved among the α-proteobacteria and dispensable for PopZ localization control, the membrane-anchored ZitP ZnR and PopZ binding is featured by many ZitP orthologs from the branch of obligate intracellular α-proteobacteria (the Rickettsiae) and regulates PopZ subcellular positioning.

The NMR solution structure of the ZitP ZnR revealed an atypical ββαββ-architecture, resembling that of the crenarchaeal DNA-binding protein Cren7 (*Zhang et al., 2010*). However, the ZitP ZnR lacks a fifth β-strand (β5) that is used by Cren7 to establish critical sequence-specific contacts with DNA. An unexpected finding was that PopZ-bound ZitP associates indirectly with sites flanking the ParB-bound centromere (*parS*), rather than *parS* itself that ParB binds (*Figure 4—figure supplement 1*). We found that the interaction between PopZ and ParAB (*Bowman et al., 2008*; *Ebersbach et al., 2008*; *Laloux and Jacobs-Wagner, 2013*; *Ptacin et al., 2014, 2010*), or at least the N-terminal region of PopZ, is required for ZitP to associate with these *parS*-flanking sites. It is also possible that another (unknown) component(s) of the PopZ complex influences the association of ZitP at these sites or that it underlies an unprecedented mode of occupancy of the macromolecular ZitP•PopZ•ParB complex on the chromosome in proximity to *parS*. Interestingly, the ParB-centromeric complex was recently proposed to self-assemble via nucleation and 'caging' in vivo, i.e. three-dimensional stochastic interactions between ParB and *parS,* and possibly reinforced or delimited by adjacent nucleoprotein complexes (*Sanchez et al., 2015*) or accessory factors that may allow supercomplexes to extend into neighboring chromosomal regions.

With the juxtaposition of PopZ•ZitP at the ParB-bound centromere and our reconstitution of the ZitP•PopZ•ParB tripartite complex in *E. coli*, ZitP is well positioned to exert control of ParB•*parS* dynamics and to impair segregation by sequestering the adjacent centromere into aberrant PopZ•ZitP clusters. The ParB-control function of PopZ•ZitP depends on the ability of membrane-anchored ZitP ZnR to interact with PopZ. Our molecular dissections provide evidence that ZitP

targets residues flanking the conserved C-terminal DUF2497 of PopZ. Thus, the DUF2497 region may serve as a key determinant in PopZ to regulate its association with (known and unknown) client proteins and its bipolar localization and/or dynamics.

The bipolar localization of DivIVA of the Gram-positive Firmicutes has been intensively studied and is governed by the biophysical properties of DivIVA that attract it to concavely curved membranes and thus both cell poles when expressed in *E. coli* (*Lenarcic et al., 2009*; *Ramamurthi and Losick, 2009*; *Strahl and Hamoen, 2012*). The α-proteobacterial PopZ system is of distinct ancestry and structural features compared to DivIVA, and it aggregates into (sporadic) monopolar and non-polar (cytoplasmic) clusters in (filamentous) *E. coli* in the absence of ZitP. While these PopZ clusters do not rely on membrane curvature to find the *E. coli* poles, their localization mechanism is not understood (*Laloux and Jacobs-Wagner, 2013*) and could reflect aggregation in a partially folded form into cellular inclusion bodies (*Miller et al., 2015*). A clearly different mechanism underlies the robust assembly of (near native) bipolar PopZ•ZitP complexes at the polar membrane as observed by PALM. It is possible that these PopZ•ZitP complexes scout the poles by a similar mechanism as DivIVA, a possibility that warrants further investigation. As these bipolar PopZ•ZitP complexes in *E. coli* can also be reconstituted with components from different α-proteobacteria, it is unlikely that it is mediated by specialized phospholipids as the composition differs between *E. coli*, *Caulobacterales* (*De Siervo, 1985*; *De Siervo and Homola, 1980*) and other α-proteobacteria, for example the Rickettsiales.

There are intriguing evolutionary implications based on the finding that rickettsial ZitP retains the aforementioned PopZ control activities. It suggests that these obligate intracellular pathogens also feature a polarized PopZ•ZitP complex. The rickettsial actin filament organizer Sca2 is polarized and promotes actin-mediated motility of some *Rickettsiae* inside the host cell cytoplasm (*Haglund et al., 2010*; *Madasu et al., 2013*). It is therefore conceivable that the PopZ•ZitP bipolarity system served as a primordial platform (*Figure 6H*) for the acquisition of specialized polarity based functions conferred via PopZ-interacting proteins that reinforce bipolarity and/or confer other polarized traits. For example, PopZ is also required for stalk biogenesis in *C. crescentus* (*Bowman et al., 2010*), a structure that requires specialized peptidoglycan (PG) biosynthesis enzymes (*Kühn et al., 2010*). As several Rhizobiales members also sequester PopZ to the site of PG synthesis to promote unipolar growth (*Anderson-Furgeson et al., 2016*; *Brown et al., 2012*; *Curtis and Brun, 2014*; *Grangeon et al., 2015*), the current challenge is to determine which proteins interact with PopZ in the different α-proteobacterial branches, particularly in the Rhizobiales. PopZ is unipolar in this branch (*Deghelt et al., 2014*; *Grangeon et al., 2015*) and no ZitP orthologs are encoded, suggesting that unknown PopZ-control mechanisms exist (*Figure 6H*,*Figure 1—figure supplement 1A*). By contrast, many members of the Rhodobacterales, do not encode a conspicuous PopZ ortholog in their genome. This, along with the structural divergence of the ZitP$^{ZnR}$ in this lineage (*Figure 3C*), suggests that ZitP has been appropriated for other functions or interactions with partners differing in primary structure.

Proper control of polarization is critical for efficient cellular proliferation and fitness in many cell types. In yeast, aberrant or misregulated polarity complexes impair the fitness of yeast cells to the extent where it is favorable to cells to eliminate these functions completely than retaining them improperly regulated (*Laan et al., 2015*). Similar selective forces must have ensured the evolution and retention of related polarity control modules in free-living and obligate intracellular bacteria since misregulation of polarity similarly perturbs the α-proteobacterial cell division cycle.

## Materials and methods

### Strains, plasmids and oligos

Strains, plasmids and oligos are listed in *Supplementary file 2*. Plasmids expressing Dendra2 variants were used (*Holden et al., 2014*) and strains are derivatives of *Caulobacter crescentus* NA1000 whose genome is sequenced (*Marks et al., 2010*).

### Growth conditions

*Caulobacter crescentus* NA1000 (*Marks et al., 2010*) and derivatives were cultivated at 30°C in peptone yeast extract (PYE)-rich medium (2 g/L bactopeptone, 1 g/L yeast extract, 1 mM MgSO$_4$, and

0.5 mM CaCl$_2$) or in M2 minimal salts (M2G) (0.87 g/L Na$_2$HPO$_4$, 0.54 g/L KH$_2$PO$_4$, 0.50 g/L NH$_4$Cl, 0.2% [wt/vol] glucose, 0.5 mM MgSO$_4$, 0.5 mM CaCl$_2$, and 0.01 mM FeSO$_4$) (*Ely, 1991*). *E. coli* strains were grown at 37°C in Luria Bertani (LB)–rich medium (10 g/L NaCl, 5 g/L yeast extract, and 10 g/L tryptone). When appropriate, media were supplemented with antibiotics at the following concentrations (µg/mL in liquid/solid medium for *C. crescentus* strains; µg/mL in liquid/solid medium for *E. coli* strains): kanamycin (5/20 µg.mL$^{-1}$; 20/20 µg.mL$^{-1}$), tetracycline (1/1 µg mL$^{-1}$; not appropriate), spectinomycin and streptomycin (in solid for *C. crescentus* only) (25/25, five respectively; 30/90 µg.mL$^{-1}$), gentamycin (1/1; 10/25 µg.mL$^{-1}$) and nalidixic acid (in solid only) (20 µg.mL$^{-1}$). When needed, for *Caulobacter*, D-xylose or sucrose was added at 0.3% final concentration, glucose at 0.2% final concentration and vanillate at 500 µM final concentration. When needed, for *E. coli*, arabinose and IPTG were added at a final concentration of 0.3% and 1 mM, respectively.

Swarmer cell isolation, electroporation, biparental mating (intergeneric conjugations) and bacteriophage φCr30-mediated generalized transduction were performed as described (*Ely, 1991*). Briefly, swarmer cells were isolated by Ludox or Percoll density-gradient centrifugation at 4°C, followed by three washes and final re-suspension in pre-warmed (30°C) M2G. Electroporation was done from 1 ml overnight cells that had been washed three times in sterile water. Biparental matings were done using exponential phase *E. coli* S17-1 donor cells and *C. crescentus* recipient cells washed in PYE and mixed at 1:10 ratio on a PYE plate. After 4–5 hr of incubation at 30°C, the mixture of cells was plated on PYE harbouring nalidixic acid (to counter select *E. coli*) and the antibiotic that the conjugated plasmid confers resistance to. Generalized transductions were done by mixing 50 µL ultraviolet-inactivated transducing lysate with 500 µL stationary phase recipient cells, incubation for 2 hr, followed by plating on PYE containing antibiotic to select for the transduced DNA.

## ZitP purification and production of antibodies

ZitP N-TER or C-TER recombinant protein, comprising only the first 90 amino acids or lacking the last 119 residues respectively, was expressed from pET28a in *E. coli* BL21(DE3)/ pLysS and purified under native conditions using Ni$^{2+}$ chelate chromatography. Cells were grown in LB at 37°C to an OD$_{600nm}$ of 0.6 and induced by the addition of IPTG to 1 mM during 3 hr, and harvested at 5000 RPM at 4°C during 30 min. Cells were pelleted and re-suspended in 25 mL of lysis buffer (10 mM Tris HCl (pH 8), 0.1 M NaCl, 1.0 M β-mercaptoethanol, 5% glycerol, 0.5 mM imidazole Triton X-100 0.02%). Cells were sonicated in a water–ice bath, 15 cycles 30 s ON; 30 s OFF. After centrifugation at 5000g for 20 min at 4°C, the supernatant was loaded onto a column containing 5 mL of Ni-NTA agarose resin (Qiagen, Hilden, Germany) pre-equilibrated with lysis buffer. The column was rinsed with lysis buffer, 400 mM NaCl and 10 mM imidazole, both prepared in lysis buffer. Fractions were collected (in 300 mM Imidazole buffer, prepared in lysis buffer) and used to immunize New Zealand white rabbits (Josman LLC).

## Fractionation

Fifty mL of an exponential culture of *Caulobacter* (OD$_{600nm}$ = 0.4) was harvested by centrifugation for 15 min at 8000g at 4°C. Cell pellets were re-suspended in 1 mL of lysis buffer (20 mM Tris-HCl pH 7.5, 300 mM NaCL, 0.5 mM EDTA, 5 mM MgCl$_2$ at 4°C freshly supplemented with 1 mM DTT, 12500 U ready-lyse (Epicentre technologies), and one tablet of EDTA-free protease inhibitor cocktail ([Complete; Roche] per 50 mL). Ten µL of DNAse 1 mg.mL$^{-1}$, 5 µL of RNAseA 20 mg.mL$^{-1}$ were added before sonication in an ice-water bath, 15 cycles 30 s ON; 30 s OFF. Twenty µL of this was mixed with 20 µL of loading buffer 2X (0.25 M Tris pH 6.8, 6% (wt/vol) SDS, 10 mM EDTA, 20% (vol/vol) Glycerol) containing 10% (vol/vol) β-mercaptoethanol to obtain the crude extract (CE). Sonicated samples were centrifuged for 30 min at 20,000g at 4°C, and the supernatant was diluted in 2X loading buffer to obtain the soluble fraction (S). The pellet, containing the insoluble fraction, was resuspended in 1 mL resuspension buffer (20 mM Tris-HCl pH 7.5, 300 mM NaCl, 5 mM EDTA, 1 mM DTT) and 20 µL was diluted in loading buffer 2X to obtain the insoluble fraction (P). The membrane fraction was split in 3 fractions of 300 µL to add 300 µL re-suspension buffer (control) or 300 µL 4M NaCl buffer (20 mM Tris-HCl pH 7.5, 4 M NaCl, 5 mM EDTA, 1 mM DTT) (solubilize proteins associated with membrane) or 300 µL 2% Triton-X100 (20 mM Tris-HCl pH 7.5, 300 mM NaCl, 5 mM EDTA, 1 mM DTT, 2% Triton-X100) (solubilize integral membrane proteins). Samples were incubated 1 hr with shaking at 4°C and harvested by centrifugation at 20,000g during 30 min at 4°C. Forty µL

of the supernatant was diluted in 40 µL loading buffer 2X to obtain the soluble fraction (S). The pellet, containing the insoluble fraction, was resuspended in 600 µl of resuspension buffer. Forty µL of this was diluted in 40 µL loading buffer 2X to obtain the insoluble fraction (P) All the fraction were analysed by immunoblot using antibodies to ZitP (NTER), DivJ (as control of integral membrane protein) and CtrA (as control of soluble proteins).

## Whole-cell extracts preparation

Five hundred µL of an exponential *Caulobacter* or *E. coli* cells ($OD_{600nm}$ = 0.4 and 0.8 respectively) were harvested with 20,000g at room temperature for 5 min. Whole-cell extracts were prepared by resuspension of cell pellets in 75 µL TE buffer (10 mM Tris-HCl pH 8.0 and 1 mM EDTA) followed by addition of 75 µL loading buffer 2X (0.25 M Tris pH 6.8, 6% (wt/vol) SDS, 10 mM EDTA, 20% (vol/vol) Glycerol) containing 10% (vol/vol) β-mercaptoethanol. Samples were normalized for equivalent loading using $OD_{600nm}$ and were heated for 10 min at 90°C prior to loading.

## Immunoblot analysis

Protein samples were separated by SDS–polyacrylamide gel electrophoresis and blotted on polyvinylidenfluoride membranes (Merck Millipore). Membranes were blocked overnight with Tris-buffered saline 1X (TBS) (50 mM Tris-HCl, 150 mM NaCl, pH 8) containing, 0.1% Tween-20 and 8% dry milk and then incubated for an additional 3 hr with the primary antibodies diluted in TBS 1X, 0.1% Tween-20, 5% dry milk. The different polyclonal antisera to ZitP (NTER, 1:5000 dilution and CTER, 1:5000), CtrA (1:10,000) and DivJ (1:10,000) were used. Commercial and polyclonal antibodies to Dendra2 (Antibodies-Online) and mCherry (*Chen et al., 2005*) were used at 1:5,000 and 1:10,000 dilutions respectively. Primary antibodies were detected using HRP-conjugated donkey anti-rabbit antibody (Jackson ImmunoResearch) with ECL Western Blotting Detection System (GE Healthcare) and a luminescent image analyzer (Chemidoc™ MP, Biorad).

## ChIP-SEQ

Mid-log phase cells were cross-linked in 10 mM sodium phosphate (pH 7.6) and 1% formaldehyde at room temperature for 10 min and on ice for 30 min thereafter, washed three times in phosphate-buffered saline (PBS) and lysed in a Ready-Lyse lysozyme solution (Epicentre Technologies) according to the manufacturer's instructions. Lysates were sonicated in a ice-water bath, 15 cycles 30 s ON; 30 s OFF to shear DNA fragments to an average length of 0.3–0.5 kbp and cleared by centrifugation at 14,000 g for 2 min at 4°C. Lysates were normalized by protein content, diluted to 1 mL using ChIP buffer (0.01% SDS, 1.1% Triton X-100, 1.2 mM EDTA, 16.7 mM Tris-HCl (pH 8.1), 167 mM NaCl plus protease inhibitors (Roche, Switzerland) and pre-cleared with 80 µl of protein-A agarose (Roche) and 100 µg BSA. To immunoprecipate the chromatin, two µL of polyclonal antibodies were added to the supernatant, incubated overnight at 4°C with 80 µL of protein-A agarose beads pre-saturated with BSA. Antibodies to ZitP NTER, ZitP CTER and ParB (*Mohl and Gober, 1997*) were used for ChIP. The immunoprecipitate was washed once with low salt buffer (0.1% SDS, 1% Triton X-100, 2 mM EDTA, 20 mM Tris-HCl (pH 8.1) and 150 mM NaCl), high salt buffer (0.1% SDS, 1% Triton X-100, 2 mM EDTA, 20 mM Tris-HCl (pH 8.1) and 500 mM NaCl) and LiCl buffer (0.25 M LiCl, 1% NP-40, 1% sodium deoxycholate, 1 mM EDTA and 10 mM Tris-HCl (pH 8.1)), and twice with TE buffer (10 mM Tris-HCl (pH 8.1) and 1 mM EDTA). The protein DNA complexes were eluted in 500 µL freshly prepared elution buffer (1% SDS and 0.1 M NaHCO3), supplemented with NaCl to a final concentration of 300 mM and incubated overnight at 65°C to reverse the crosslinks. The samples were treated with 2 µg of Proteinase K for 2 hr at 45°C in 40 mM EDTA and 40 mM Tris-HCl (pH 6.5). DNA was extracted using phenol:chloroform:isoamyl alcohol (25:24:1), ethanol precipitated using 20 µg of glycogen as carrier and resuspended in 100 µL of water.

Immunoprecipitated chromatin was used to prepare sample libraries used for deep-sequencing at Fasteris SA (Geneva, Switzerland). ChIP-Seq libraries were prepared using the DNA Sample Prep Kit (Illumina) following manufacturer's instructions. Single-end run were performed on an Illumina Genome Analyzer IIx or HiSeq2000, 50 cycles were read and yielded several million reads. The single-end sequence reads stored in FastQ files were mapped against the genome of *Caulobacter crescentus* NA1000 (NC_011916) and converted to SAM using BWA and SAM tools respectively from the galaxy servor (https://usegalaxy.org/). The resulting SAM was imported into SeqMonk (http://

www.bioinformatics.babraham.ac.uk/projects/seqmonk/, version 0.21.0) to build sequence read profiles. The initial quantification of the sequencing data was done in SeqMonk: the genome was subdivided into 50 bp probes, and for every probe we calculated a value that represents a normalized read number per million. All ChIP-seq data was deposited in the GEO database under accession number GSE79918 (https://www.ncbi.nlm.nih.gov/geo/query/acc.cgi?acc=GSE79918).

## Isothermal titration calorimetry (ITC)

ITC experiments were performed on a VP-ITC instrument (Microcal). Both partners were prepared in the same NMR spectroscopy buffer, ZitP at 0.015 mM and PopZ at 0.300 mM. ZitP was titrated using a solution of PopZ by 65 injections of 4 µL every 300 s at 20°C. The raw data were integrated, normalized for the molar concentration and analyzed using Origin7.0 according to a 1:1 binding model.

## Growth measurements in broth

The overnight cultures were started in PYE supplemented with D-xylose 0.3% final. The cultures were diluted to obtain an $OD_{600nm}$ of 0.1 in PYE supplemented with D-xylose 0.3% final. The $OD_{600nm}$ was recorded every hour during 9 hr. The graph represents the trend of the growth curve of three independent experiments.

## Plasmid constructions

### pMB123 (pNTPS138- ΔzitP)

The plasmid construct used to delete *zitP* (*CCNA_02298*) was made by PCR amplification of two fragments. The first, a 771 bp fragment was amplified using primers OMB87 and OMB88 (table S?), flanked by an *Eco*RI and a *Bam*HI site. The second, a 889 bp fragment was amplified using primers OMB89 and OMB90, flanked by a *Bam*HI site and a *Hind*III site. These two fragments were first digested with appropriate restriction enzymes and then triple ligated into pNTPS138 (M.R.K. Alley, Imperial College London, unpublished) previously restricted with *Eco*RI/*Hind*III.

### pXdendra2-N2

We PCR amplified the photoactivatable variant *dendra2* from the pX-*ftsZ-dendra2* (*Biteen et al., 2012*) with dendra2N_F and dendra2N _R primers. This fragment was digested with *Nde*I/*Pac*I and cloned into *Nde*I/*Pac*I-digested pMT582 (*Thanbichler et al., 2007*).

### pXdendra2-N2-*zitP*$^{WT}$

The *zitP*-coding sequence (nt 2444803–2445738, NA1000 genome) was PCR amplified from the NA1000 strain using the 2215sh + 2_SacI and CC2215_E primers. This fragment was digested with *Sac*I/*Eco*RI and cloned into *Sac*I/*Eco*RI-digested pX-*dendra2*N2

### pMB22 (pXdendra2-N2-*zitP*$^{1-90(Cc)}$)

The first 270nt of the ZitP coding sequence were PCR amplified from the NA1000 strain using the OMB12 and OMB13 primers. This fragment was digested with *Sac*I/*Eco*RI and cloned into *Sac*I/*Eco*RI-digested pXdendra2-N2 (Seamus Holden, unpublished).

### pMB33 (pXdendra2-N2-zitP$^{1-80(Ae)}$)

The first 240 nt of the Astex_3267 coding sequence were PCR amplified from the synthetic fragment one using the Van and T7pro primers. This fragment was digested with *Sac*I/*Eco*RI and cloned into *Sac*I/*Eco*RI-digested pXdendra2-N2.

### pMB38 (pXdendra2-N2-zitP$^{1-87(Mm)}$)

The first 261 nt of the Mmar10_2373 coding sequence were PCR amplified from the synthetic fragment two using the Van and T7pro primers. This fragment was digested with *Sac*I/*Eco*RI and cloned into *Sac*I/*Eco*RI-digested pXdendra2-N2.

### pMB39 (pXdendra2-N2-zitP$^{1-89(Cs)}$)

The first 267 nt of the *Cseg_28671* coding sequence were PCR amplified from the synthetic fragment three using the Van and T7pro primers. This fragment was digested with *SacI/EcoRI* and cloned into *SacI/EcoRI*-digested pXdendra2-N2.

### pMB26 (pXdendra2-N2-$^{agmX1-90}$)

The 90 first amino acid of AgmX coding sequence were PCR amplified from the synthetic fragment 20 using Van and T7pro primers. This fragment was digested with *SacI/EcoRI* and cloned into *SacI/EcoRI*-digested pXdendra2-N2.

### pMB66 (pXdendra2-N2-zitP$^{1-43WT(Cc)}$)

The first 129 nt of the ZitP coding sequence were PCR amplified from the NA1000 strain using the OMB12 and OMB41 primers. This fragment was digested with *SacI/EcoRI* and cloned into *SacI/EcoRI*-digested pXdendra2-N2.

### pMB164 (pXdendra2-N2-zitP$^{1-43CS(Cc)}$)

The first 129 nt of the ZitP coding sequence were PCR amplified from the pMT335-2215CS strain using the OMB12 and OMB41 primers. This fragment was digested with *SacI/EcoRI* and cloned into *SacI/EcoRI*-digested pXdendra2-N2.

### pMB23 (pXdendra2-N2-zitP$^{1-133}$)

The first 396 nt of the ZitP coding sequence were PCR amplified from the NA1000 strain using the OMB12 and OMB14 primers. This fragment was digested with *SacI/EcoRI* and cloned into *SacI/EcoRI*-digested pXdendra2-N2.

### pMB25 (pXdendra2-N2-zitP$^{1-133CS}$)

The *zitP$^{CS}$* tetracysteine mutant allele was PCR amplified from the pMT335-2215CS using the 2215shCS + 2_SacI and OMB14 primers. This fragment was digested with *SacI/EcoRI* and cloned into *SacI/EcoRI*-digested pX-dendra2N2.

### pMB73 (pXdendra2-N2-zitP$^{1-49(Bd)}$)

The first 147 nt of HMPREF0185_02600 coding sequence were PCR amplified from the synthetic fragment four using the xyl and lac290 primers. This fragment was digested with *SacI/EcoRI* and cloned into *SacI/EcoRI*-digested pXdendra2-N2.

### pMB115 (pXdendra2-N2-zitP$^{1-43W35I(Cc)}$)

The first 129 nt of the ZitP coding sequence were PCR amplified from the synthetic fragment 5 (containing the W35I mutation) using the T7pro and T7ter primers. This fragment was digested with *SacI/EcoRI* and cloned into *SacI/EcoRI*-digested pXdendra2-N2.

### pMB170 (pXdendra2-N2-zitP$^{1-43R24A/R27A(Cc)}$)

The first 129 nt of the ZitP coding sequence were PCR amplified from the synthetic fragment 6 (containing the R24A/R27A mutation) using the van and T7pro primers. This fragment was digested with *SacI/EcoRI* and cloned into *SacI/EcoRI*-digested pXdendra2-N2.

### pMB184 (pXdendra2-N2-zitP$^{1-43WT(Rs)}$)

The first 129 nt of the Rsph17025_2450 coding sequence were PCR amplified from the synthetic fragment seven using the van and xylAS primers. This fragment was digested with *SacI/EcoRI* and cloned into *SacI/EcoRI*-digested pXdendra2-N2.

### pMB91 (pXdendra2-N2-zitP$^{1-43Q27R(Rs)}$)

The first 129 nt of the Rsph17025_2450 coding sequence were PCR amplified from the synthetic fragment eight using the van and xylAS primers. This fragment was digested with *SacI/EcoRI* and cloned into *SacI/EcoRI*-digested pXdendra2-N2.

### pMB185 (pXdendra2-N2-$zitP^{1-43Q27R/W35I(Rs)}$)

The first 129 nt of the Rsph17025_2450 coding sequence were PCR amplified from the synthetic fragment 18 using the van and T7pro primers. This fragment was digested with SacI/EcoRI and cloned into SacI/EcoRI-digested pXdendra2-N2.

### pMB81 (pXdendra2-N2-$zitP^{1-50WT(Rm)}$)

The first 150 nt of RMB_01390 coding sequence were PCR amplified from the synthetic fragment nine using the van and xylAS primers. This fragment was digested with SacI/EcoRI and cloned into SacI/EcoRI-digested pXdendra2-N2.

### pMB116 (pXdendra2-N2-$zitP^{1-43W35I(Rm)}$)

The first 150 nt RMB_01390 coding sequence were PCR amplified from the synthetic fragment 10 using the van and xylAS primers. This fragment was digested with SacI/EcoRI and cloned into SacI/EcoRI-digested pXdendra2-N2.

### pMT335-$zitP^{CS:}$

We PCR amplified the $zitP^{CS}$ tetracysteine mutant allele from pMT335-$2215^{CS}$ with 2215shCS_NdeI and CC2215_E primers. This fragment was digested with NdeI/EcoRI and cloned into NdeI/EcoRI-digested pMT335 (*Thanbichler et al., 2007*).

### pMB21 (pMT335-$popZ$)

The PopZ coding sequence was PCR amplified from NA1000 using the OMB7 and OMB11 primers. This fragment was digested with NdeI/EcoRI and cloned into NdeI/EcoRI-digested pMT335.

### pMB93 (pMT464-$zitP^{1-132WT(Cc)}$)

The first 396 nt of ZitP coding sequence was PCR amplified from NA1000 using the 2215sh_NdeI and OMB14 primers. This fragment was digested with NdeI/EcoRI and cloned into NdeI/EcoRI-digested pMT464.

### pMB94 (pMT464-$zitP^{1-132CS(Cc)}$)

The first 396nt of $zitP$ sequence was PCR amplified from fragment 11 (containing the cysteines C5, C8, C28 and C31 mutated in serine) using the Van and OMB14 primers. This fragment was digested with NdeI/EcoRI and cloned into NdeI/EcoRI-digested pMT464.

### pMB113 (pMT464-$zitP^{1-132W35I(Cc)}$)

The first 396nt of $zitP$ sequence was PCR amplified from fragment 12 (containing the mutation W35I) using the Van and T7 primers. This fragment was digested with NdeI/EcoRI and cloned into NdeI/EcoRI-digested pMT464.

### pMB127 (pMT464-$zitP^{1-132malF}$)

The first 396 nt of the ZitP coding sequence was PCR amplified from fragment 13 (containing the MalF transmembrane domain instead the natural transmembrane domain from ZitP) using the Van and T7 primers. This fragment was digested with NdeI/EcoRI and cloned into NdeI/EcoRI-digested pMT464.

### pMB156 (pMT464-$zitP^{1-120WT(Bd)}$)

The first 360 nt of the HMPREF0185_02600 coding sequence was PCR amplified from fragment 14 (containing the ZnR from *B. diminuta* and its transmembrane domain) using the Van and T7 primers. This fragment was digested with NdeI/EcoRI and cloned into NdeI/EcoRI-digested pMT464.

### pMB114 (pMT464-$zitP^{1-104WT(Rm)}$)

The first 312nt of the ZitP coding sequence was PCR amplified from fragment 15 (containing the ZnR from *R. massiliae* and its transmembrane domain) using the Van and T7 primers. This fragment was digested with NdeI/EcoRI and cloned into NdeI/EcoRI-digested pMT464.

## pMB145 (pMCS-4-popZ$^{\Delta26}$)

The PopZ coding sequence deleted for the 26 first amino acids was PCR amplified from NA1000 using the OMB70 and OMB11 primers. This fragment was digested with KpnI/EcoRI and cloned into KpnI/EcoRI-digested pJP312 (**Ptacin et al., 2014**).

## pMB49 (pBAD101-popZ)

The popZ coding sequence was PCR amplified from pMB21 using the OMB31 and M13(−20) primers. These fragments were digested with NcoI/NdeI and NdeI/XbaI respectively and triple ligated into NcoI/XbaI-digested pBAD101.

## pMB59 (pBAD101-mCherry-popZ)

The mCherry coding sequence was PCR amplified using the OMB35 and OMB36 and popZ was PCR amplified from pMB21 using the van and M13(−20) primers. These fragments were digested with NcoI/NdeI and NdeI/XbaI respectively and triple ligated into NcoI/XbaI-digested pBAD101.

## pMB128 (pBAD101-mCherry-popZ$^{\Delta26}$)

The mCherry coding sequence was PCR amplified using the OMB35 and OMB36 and popZ was PCR amplified from NA1000 using the OMB65 and OMB66 primers. These fragments were digested with NcoI/NdeI and NdeI/XbaI respectively and triple ligated into NcoI/XbaI-digested pBAD101.

## (pSRK-dendra2-zitP$^{1-43WT(Cc)}$)

The ZitP$^{1-43}$ coding sequence was PCR amplified from pMB66 using OMB76 and M13(−20) primers. This fragment was digested with SacI /XbaI and ligated into SacI/XbaI-digested pMB86.

## pMB193 (pSRK-dendra2-zitP$^{1-43CS(Cc)}$)

The ZitP$^{1-43CS}$ coding sequence was PCR amplified from pMB164 using OMB76 and M13(−20) primers. This fragment was digested with SacI /XbaI and ligated into SacI/XbaI-digested pMB86.

## pMB194 (pSRK-dendra2-zitP$^{1-43R24A/R27A(Cc)}$)

The ZitP$^{1-43R24A/R27A}$ coding sequence was PCR amplified from pMB170 using OMB76 and M13(−20) primers. This fragment was digested with SacI /XbaI and ligated into SacI/XbaI-digested pMB86.

## pMB195 (pSRK-dendra2-zitP$^{1-43WT(Rs)}$)

The ZitP$^{1-43WT(Rs)}$ coding sequence was PCR amplified from pMB184 using OMB76 and M13(−20) primers. This fragment was digested with SacI /XbaI and ligated into SacI/XbaI-digested pMB86.

## pMB196 (pSRK-dendra2-zitP$^{1-43Q27R(Rs)}$)

The ZitP$^{1-43Q27R(Rs)}$ coding sequence was PCR amplified from pMB91 using OMB76 and M13(−20) primers. This fragment was digested with SacI /XbaI and ligated into SacI/XbaI-digested pMB86.

## pMB197 (pSRK-dendra2-zitP$^{1-43Q27R/W35I(Rs)}$)

The ZitP$^{1-43Q27R/W35I(Rs)}$ coding sequence was PCR amplified from pMB185 using OMB76 and M13(−20) primers. This fragment was digested with SacI /XbaI and ligated into SacI/XbaI-digested pMB86.

## pMB166 (pSRK-dendra2-zitP$^{1-49WT(Bd)}$)

The HMPREF0185_02600$^{1-49}$ coding sequence was PCR amplified from pMB73 using OMB76 and M13(−20) primers. This fragment was digested with SacI/XbaI and ligated into SacI/XbaI-digested pMB86.

### pMB167 (pSRK-*dendra2-zitP2373*$^{1-87WT(Mm)}$)

The Mmar10_2373$^{1-87}$ coding sequence was PCR amplified from pMB38 using OMB76 and M13 (−20) primers. This fragment was digested with *Sac*II/*Xba*I and ligated into *Sac*II/*Xba*I-digested pMB86.

### pMB97 (pSRK-*dendra2-zitP*$^{1-50WT(Rm)}$)

The RMB_01380$^{1-50WT}$ coding sequence was PCR amplified from pMB81 using OMB76 and M13 (−20) primers. This fragment was digested with *Sac*I/*Xba*I and ligated into *Sac*I/*Xba*I-digested pMB86.

### pMB168 (pSRK-*dendra2-zitP*$^{1-50W35I(Rm)}$)

The RMB_01380$^{1-50W35I}$ coding sequence PCR amplified from pMB116 using OMB76 and M13(−20) primers. This fragment was digested with *Sac*I/*Xba*I and ligated into *Sac*I/*Xba*I-digested pMB86.

### pMB169 (pSRK-*dendra2-zitP*$^{1-43W35I(Cc)}$)

The ZitP$^{1-43W35I}$ coding sequence was PCR amplified from pMB115 using OMB76 and M13(−20) primers. This fragment was digested with *Sac*I/*Xba*I and ligated into *Sac*I/*Xba*I-digested pMB86.

### pMB43 (pSRK-*dendra2-zitP*$^{WT}$)

The Dendra2-ZitP coding sequence was PCR amplified from p*X-dendra2-zitP*$^{WT}$ using xylseq2 and M13(−20) primers. This fragment was digested with *Nde*I/*Xba*I and ligated into *Nde*I/*Xba*I-digested pSRK.

### pMB45 (pSRK-*dendra2-zitP*$^{1-90}$)

The Dendra2-ZitP$^{1-90}$ coding sequence was PCR amplified from p*X-dendra2-zitP*$^{WT}$ using xylseq2 and OMB13 primers. This fragment was digested with *Nde*I/*Xba*I and ligated into *Nde*I/*Xba*I-digested pSRK.

### pMB86 (pSRK-*dendra2-zitP*$^{1-133WT(Cc)}$)

The Dendra2-ZitP$^{1-133WT}$ coding sequence was PCR amplified from pMB23 using xylseq2 and M13 (−20) primers. This fragment was digested with *Nde*I/*Xba*I and ligated into *Nde*I/*Xba*I-digested pSRK.

### pMB178 (pSRK-*dendra2-zitP*$^{1-133W35I(Cc)}$)

The ZitP$^{1-133W35I}$ coding sequence was PCR amplified from pMB113 using OMB80 and OMB84 primers. This fragment was digested with *Sac*I/*Xba*I and ligated into *Sac*I/*Xba*I-digested pMB86.

### pMB179 (pSRK-*dendra2-zitP*$^{1-133malF}$)

The ZitP$^{1-133malF}$ coding sequence was PCR amplified from pMB127 using OMB81 and OMB84 primers. This fragment was digested with *Sac*I/*Xba*I and ligated into *Sac*I/*Xba*I-digested pMB86.

### pMB181 (pSRK-*dendra2-zitP*$^{1-104WT(Rm)}$)

The RMB_01390$^{1-104WT}$ coding sequence was PCR amplified from pMB114 using OMB83 and OMB84 primers. This fragment was digested with *Sac*I/*Xba*I and ligated into *Sac*I/*Xba*I-digested pMB86.

### pMB175 (pMT464-zitP$^{1-133WT}$-*popZ*)

The PopZ coding sequence was PCR amplified from NA1000 using the OMB78 and OMB79 primers adding an *Eco*RI plus an RBS and an *Xba*I site respectively. This fragment was digested with *Eco*RI/*Xba*I and cloned into *Eco*RI/*Xba*I -digested pMB93.

### pMB177 (pMT464-zitP$^{1-133W35I}$-popZ)

The PopZ coding sequence was PCR amplified from NA1000 using the OMB78 and OMB79 primers adding an EcoRI plus an RBS and an XbaI site respectively. This fragment was digested with EcoRI/XbaI and cloned into EcoRI/XbaI -digested pMB113.

### pSC

: pSC is a derivative of pET26b (Novagen) in which expression of the protein of interest is in frame with a coding sequence for C-terminal thrombin cleavage site and a His$_6$-tag. The coding sequence can be cloned between NdeI and XhoI. An NheI site was also introduced after the stop codon that enables an easy construction of polycistronic synthetic genes.

### pMB64 (pSC-popZ)

The PopZ coding sequence (codon optimized for E. coli) was PCR amplified from fragment 16 using Van and T7 primers. This fragment was digested with NdeI/XhoI and ligated into NdeI/XhoI-digested pSC.

### pMB54 (pSC-zitP$^{1-90}$)

The first 270 nt of the ZitP coding sequence (codon optimized for E. coli) was PCR amplified from fragment 17 using Van and T7 primers. This fragment was digested with NdeI/XhoI and ligated into NdeI/XhoI-digested pSC.

### pET28a-zitP$^{Cterm}$

The zitP-coding sequence lacking the first 189 bp from the start codon (nt 1082122–1082481) was PCR amplified from the NA1000 strain using the zitP_Cterm_nde and zitP_Cterm_eco primers. This fragment was digested with NdeI/EcoRI and cloned into NdeI/EcoRI-digested pET28a (Novagen).

### pSC-zitP$^{1-43WT}$

The first 129 nt of the ZitP coding sequence (codon optimized for E. coli) was PCR amplified from fragment 17 using zitP1-43_fw_NdeI and zitP1-43_fw_XhoI primers. This fragment was digested with NdeI/XhoI and ligated into NdeI/XhoI-digested pSC.

### pMB140 (pSC-zitP$^{1-43W35I}$)

The first 129 nt of the ZitP coding sequence (carrying the W35I mutation and codon optimized for E. coli) was PCR amplified from fragment 19 using Van and T7 primers. This fragment was digested with NdeI/XhoI and ligated into NdeI/XhoI-digested pSC.

### pMB233 (pSRK-dendra2-parB)

The dendra2 and parB coding sequence were PCR amplified from pXdendra2N2 with OMB18 and OMB134 and from NA1000 with OMB130 and OMB131 respectively. These fragment were digested with NdeI/BamHI and BamHI/HindIII and triple ligated into NdeI/HindIII-digested pSRK.

### pMB267 (pBAD22-mCherry-popZ)

The mCherry coding sequence was PCR amplified using the OMB35 and OMB36 and popZ was PCR amplified from pMB21 using the van and M13(−20) primers. These fragments were digested with NcoI/NdeI and NdeI/XbaI respectively and triple ligated into NcoI/XbaI-digested pBAD22.

### pMB224 (pBAD22-mCherry-zitP$^{1−133}$-popZ)

The zitP$^{1-133}$-popZ was digested from pMB175 using NdeI/XbaI enzyme and ligated into NdeI/XbaI-digested pMB267.

### pMB225 (pBAD22-mCherry-zitP$^{1-133(W35I)}$-popZ)

The zitP$^{1-133(W35I)}$-popZ was digested from pMB177 using NdeI/XbaI enzyme and ligated into NdeI/XbaI-digested pMB267.

## pMB255 (pMT464-zitP$^{1-90}$)

The zitP$^{1-90}$ coding sequence was PCR amplified from NA1000 using the 2215sh_ndeI and OMB13 primers. The fragment was digested with NdeI/EcoRI and ligated into NdeI/EcoRI-digested pMT464.

## pMB256 (pMT464-zitP)

The zitP coding sequence was PCR amplified from NA1000 using the 2215sh_ndeI and CC2215_E primers. The fragment was digested with NdeI/EcoRI and ligated into NdeI/EcoRI-digested pMT464.

## pMB257 (pMT464-zitP$^{1-90}$-popZ)

The zitP$^{1-90}$ coding sequence was digested from pMB255 using NdeI/EcoRI and ligated into NdeI/EcoRI-digested pMB175.

## pMB258 (pMT464-zitP-popZ)

The zitP coding sequence was digested from pMB256 using NdeI/EcoRI and ligated into NdeI/EcoRI-digested pMB175.

## pMB265 (pMT464-zitP$^{1-104(Rm)}$-popZ)

The ZnR from R. massiliae and its transmembrane domain (zitP$^{1-104(Rm)}$) coding sequence was digested from pMB114 using NdeI/EcoRI and ligated into NdeI/EcoRI-digested pMB175.

## pMB266 (pBAD22-mCherry-zitP$^{1-104(Rm)}$-popZ)

The (ZitP$^{1-104(Rm)}$)-popZ co-expression construct was digested from pMB265 using NdeI/XbaI and ligated into NdeI/XbaI-digested pMB267.

## Synthetic fragment used in this study

Fragment 1 : Astex_3267$^{1-80}$

gccgaccgactgagacgctcacaaGAGCTCTGATGCTGCTGACGTGCCCGAAGTGCGCCCTGTCGTACGC
GATCGATGGTGCGCAGCTGGGCCCCCAGGGCCGCACGGTGCGCTGCGCCAGCTGTAAGACCAC
CTGGCACGCCGAGAAGCCGGAGGAGCCGATCGAGCTGCCCCTCGAGAAGGCCGTCGAAAAGC
CCGCGACGGGCCTGAAGGAGGTGAAGGCCAAGAAGATCCCGAGCCTGTACCGCGACATGATCG
AGAGCCAGAAGCGCTGAGAATTCtatagtgagtcgtattaattt

Fragment 2 : Mmar10_2373$^{1-90}$

gccgaccgactgagacgctcacaagagctcccATGTCGATCGTCCTGTCCTGCCCCTCGTGTACCACCCGCTAT
CGCGCGAACCCGAATGCCATCGGCACGAATGGCCGCCGCGTGCGTTGCGCCTCGTGCGGCCAC
GTGTGGACCGCCGAGGTGGAAGATCCCTCGGATCTGCCGTCGCTCCAGCCGGCCCCCCCGGTC
ACGCCGGAAGCGCCGGCCGAAGAAGCGGGCGCCGAAAAGAAGGTCCACACGGCCTTCCGCGA
GCGCCAGGAGAAGAAGCGGCGTACGCTGTCCtgaattctatagtgagtcgtattaattt

Fragment 3 : Cseg_28671$^{1-90}$

gccgaccgactgagacgctcacaaGAGCTCCCATGATCCTGACGTGCCCGGAATGCGCCAGCCGTTATTTC
GTGGACGACAGCAAGGTCGGTCCGGAAGGTCGCGTCGTCCGCTGTGCCGCCTGTGGCCATCGG
TGGACGGCGCGCAATGAAGATGCCACCGATCGTTCGAAGATCCGGAAAACCCCTCGCTGGCCT
CGCGTGGTGCCGCGGATGTCGCCACGGCCAGCGCCGAAGAACCCCCCCAACCGGAAGCGGCG
GAAGAACCCCCGGTGTCGGCGCTGCCGGGTGAAGAGCTGTGAATTCtatagtgagtcgtattaattt

Fragment 4 : HMPREF0185_02600$^{1-50}$

aggatttcgcgctggtcagacaaGAGCTCTGATGATCCTGACCTGCCCGGCTTGCGCTACCTCTTACTTCGT
TCCGGACGAAGCTATCGGTCCGAACGGTCGTCGTGTTCGTTGCAAAACCTGCGGTCACGACTGG
CGTGCTTCTCTGGAAGACGCTCCGCTGGAACTGGAACCGGCTTGAATTCagctatgaccatgattacggatt

### Fragment 5 : zitP[1-43W35I]

tcaagaccggtttagaggccccaaGAGCTCTAATGATACTGACCTGCCCGGAGTGCGCCAGCCGCTATTTC
GTCGACGACTCCAAGGTCGGGCCGGACGGTCGCGTCGTGCGTTGCGCCTCTTGCGGCAATCGC
ATCACCGCCTTCAAGGACGAAGCTGAATGAattctatagtgagtcgtattaattt

### Fragment 6 : zitP[1-43R24A/R27A]

TCGTGACGTTCGTTGCTCTAACTGCGGTCACGGTATGATACTGACCTGCCCGGAGTGCGCCAGCC
GCTATTTCGTCGACGACTCCAAGGTCGGGCCGGACGGTGCCGTCGTGGCCTGCGCCTCTTGCGG
CAATCGCTGGACCGCCTTCAAGGACGAAGCTGAATGAATTCtatagtgagtcgtattaattt

### Fragment 7 : Rsph17025_2450[1-43WT]

gccgaccgactgagacgctcacaaGAGCTCTGATGCGTCTGATCTGCCCGAACTGCGACGCTCAGTACGAA
GTTTCTGACGACGCTATCCCGCCGGAAGGTCGTGACGTTCAGTGCTCTAACTGCGGTCACGGTTG
GTTCCAGCGTCCGGTTTCTCTGGCTTGAATTCaggatttcgcgctggtcagacaa

### Fragment 8 : Rsph17025_2450[1-43Q24R]

TCGTGACGTTCGTTGCTCTAACTGCGGTCACGGTATGCGTCTGATCTGCCCGAACTGCGACGCTC
AGTACGAAGTTTCTGACGACGCTATCCCGCCGGAAGGTCGTGACGTTCGTTGCTCTAACTGCGGT
CACGGTTGGTTCCAGCGTCCGGTTTCTCTGGCTTGAATTCaggatttcgcgctggtcagacaa

### Fragment 9 : RMB_01390[1-50WT]

aggatttcgcgctggtcagacaaGAGCTCTGATGTATATCACCTGCCCGAACTGCCAGACCCGTTTTATCG
TTACCTCTAACCAGATCGGCATCAACGGTCGTCGTGTTAAATGCTCTAAATGCTCCCACCTGTGG
TACCAGAAGCTGGACTACAACACCTCTACTCTGAACGACTTCAAATGAATTCagctatgaccatgattacg
gatt

### Fragment 10 : RMB_01390[1-50W35I]

gccgaccgactgagacgctcacaaGAGCTCTGATGTATATCACCTGCCCGAACTGCCAGACCCGTTTTATCG
TTACCTCTAACCAGATCGGCATCAACGGTCGTCGTGTTAAATGCTCTAAATGCTCCCACCTGATCT
ACCAGAAGCTGGACTACAACACCTCTACTCTGAACGACTTCAAATGAATTCaggatttcgcgctggtcaga
caa

### Fragment 11 : zitP[1-133CS]

gccgaccgactgagacgctcacaaAACATATGTCGATCCGCAAGGCGCGTCACTGGGCTTTCTGGACATCA
GGGTTAAAATCAGCGACCCCAGTAAGTTATGTGGGGTTCGTTCGATTCGCGGCCATGATACTGAC
CTCCCCGGAGTCCGCCAGCCGCTATTTCGTCGACGACTCCAAGGTCGGGCCGGACGGTCGCGTC
GTGCGTTCCGCCTCTTCCGGCAATCGCTGGACCGCCTTCAAGGACGAAGCTGAAGAGCTGCTCG
ACCTCTTCGAAGAGCCTGCCGCCGCCAGCGCCAGATCCCAGGGTGATCGCGACGAAGCCGCGG
AAGAAGCCGTCGCCGCCGAGGCCGAAGAGCCACCGGTCAGCGCGCTTCCGGGCGAAGAACTT
CTTtatagtgagtcgtattaattt

### Fragment 12 : zitP[1-133W35I] (codon optimized for E. coli)

gccgaccgactgagacgctcacaacatATGATCCTGACGTGCCCCGAATGCGCCTCGCGTTATTTTGTGGAT
GATTCCAAGGTCGGTCCCGATGGTCGTGTGGTCCGTTGTGCCAGCTGTGGTAATCGTATCACGGC
CTTTAAGGATGAAGCGGAAGAACTGCTGGATCTGTTTGAAGAACCGGCGGCGGCCAGCGCCCGT
AGCCAAGGTGATCGTGATGAAGCGGCGGAAGAAGCGGTGGCCGCCGAAGCGGAAGAACCGCC
GGTCAGCGCCCTGCCCGGTGAAGAACTGCCCAAGGTCTTTCGCGCGCGTGCGGATGCCGAACG
TCGTCTGCGTGCGGCGACCGCCACCGGCGTGATCTGGGCCGGTATGGCGGCCGCCATGGCGGT
CGTCGTCGTGGCGGCCCTGATCTTTCGTATCGATTGAATTCtatagtgagtcgtattaattt

### Fragment 13 : zitP[1-132malF] (codon optimized for E. coli)

gccgaccgactgagacgctcacaaCATATGATTCTGACCTGTCCGGAATGTGCATCTCGCTACTTTGTTGAC
GACAGCAAGGTTGGCCCGGACGGTCGTGTTGTACGTTGTGCGTCTTGTGGCAACCGTTGGACCG
CTTTCAAAGACGAAGCAGAAGAACTGCTGGATCTGTTTGAGGAGCCGGCTGCTGCATCCGCGCG

TAGCCAGGGCGATCGTGACGAAGCTGCGGAAGAAGCTGTAGCGGCAGAAGCGGAAGAGCCGC
CGGTTAGCGCGCTGCCGGGTGAAGAACTGCCGAAGGTGTTCCGTGCCCGCGCTGATGCGGAAC
GCCGTCTGCGTGCGGCTCTGAAATGGTCCGTACTGGGTCTGCTGGGCCTGCTGGTTGGTTACCTG
GTTGTTCTGATGTACGCCTGAATTCtctagaatatagtgagtcgtattaattt

## Fragment 14 : *HMPREF0185_02600*[1-121WT] (codon optimized for *E. coli*)

gccgaccgactgagacgctcacaaCATATGATCCTGACGTGTCCGGCGTGTGCGACCTCGTATTTTGTCCCG
GATGAAGCCATCGGTCCCAATGGTCGTCGTGTCCGTTGTAAGACCTGTGGTCATGATTGGCGTGC
CTCGCTGGAAGATGCCCCGCTGGAACTGGAACCCGCGACCGAAGGTCTGAGCCCGGCGGCCGA
TCCGGCGAGCGAAACCCTCCCCGAATCCCTGGCCGAAACCCCGGCCCCCGAACTGCCGCGTGC
CTTTCGTGCCCGTGCCGAACGTAAGCGTCGTACCCGTCAAGCTGCTGCTGCTGGTGCTGCTTGGG
CTGCTGCTGCCGTCGTTTTGGGTTTGATTACGGGTGGTGTTTTGTTTCGTGAAGAATGAATTC
tatagtgagtcgtattaattt

## Fragment 15 : *RMB_01390*[1-102WT] (codon optimized for *E. coli*)

gccgaccgactgagacgctcacaaCATATGTACATCACCTGTCCGAACTGCCAGACCCGCTTCATCGTCAC
GTCGAATCAGATCGGCATCAACGGCCGTCGCGTGAAGTGCTCGAAGTGCTCCCATCTGTGGTAT
CAGAAGCTCGATTACAATACGAGCACCCTGAACGACTTCAAGGACAAGGTCAACACGGGCACC
ATCAAGACGCCGATCAAGAATCACTATAACGCGAACGTCCCGGTCATCCTGCCGTACATGCCCC
CGAAGAAGAAGTATAACATCTTTCCGATCCTGTGGACCTCGTTCATCATCTTCTGCCTCGTGATCC
TGCTGATCGACTGAattctatagtgagtcgtattaattt

## Fragment 16 : *popZ* (codon optimized for *E. coli*)

gccgaccgactgagacgctcacaacatCATATGTCTGATCAAAGCCAAGAGCCCACGATGGAGGAAATCCTT
GCGAGCATTCGTCGCATCATCTCTGAAGATGATGCCCCCGCGGAACCAGCCGCTGAAGCGGCCC
CACCTCCTCCTCCTGAGCCGGAACCTGAACCGGTTTCGTTTGACGACGAAGTGCTGGAACTGACT
GACCCGATTGCACCTGAACCGGAGCTGCCGCCGCTTGAAACCGTAGGCGACATTGATGTTTACT
CTCCCCCGGAACCGGAATCTGAACCGGCGTATACCCCGCCGCCTGCGGCTCCTGTGTTCGACCG
TGATGAAGTGGCGGAACAACTGGTTGGCGTATCAGCTGCGTCTGCAGCGGCGAGTGCGTTTGGC
TCTTTGTCGAGTGCATTACTGATGCCTAAGGACGGCCGTACCTTAGAAGACGTAGTGCGCGAACT
GCTTCGCCCCCTGCTGAAAGAATGGCTGGATCAGAACTTGCCGCGTATTGTGGAGACGAAAGTT
GAAGAAGAAGTCCAGCGCATTTCTCGCGGCCGTGGCGCCCTCGAGtatagtgagtcgtattaattt

## Fragment 17 : *zitP*[1-90] (codon optimized for *E. coli*)

gccgaccgactgagacgctcacaacatCATATGATCCTGACTTGTCCGGAATGTGCCAGCCGTTATTTCGTCG
ACGACTCTAAAGTTGGTCCGGATGGCCGTGTTGTCCGTTGCGCGTCTTGCGGTAACCGCTGGAC
GGCATTCAAAGATGAAGCTGAAGAACTGCTGGATCTGTTTGAGGAACCGGCTGCTGCTTCTGCAC
GTTCCCAAGGTGATCGTGATGAAGCGGCGGAAGAGGCCGTTGCTGCAGAAGCCGAAGAGCCGC
CGGTTTCTGCTCTGCCAGGCGAAGAACTGCCGAAACTCGAGtatagtgagtcgtattaattt

## Fragment 18 : *zitP*[1-43R27Q/W35I(Rs)]

gccgaccgactgagacgctcacaacatGAGCTCTGATGCGTCTGATCTGCCCGAACTGCGACGCTCAGTACG
AAGTTTCTGACGACGCTATCCCGCCGGAAGGTCGTGACGTTCGTTGCTCTAACTGCGGTCACGGT
atcTTCCAGCGTCCGGTTTCTCTGGCTTGAGAATTCtatagtgagtcgtattaattt

## Fragment 19 : *zitP*[1-43W35I] (codon optimized for *E. coli*)

gccgaccgactgagacgctcacaaCATATGATCCTGACTTGTCCGGAATGTGCCAGCCGTTATTTCGTCGAC
GACTCTAAAGTTGGTCCGGATGGCCGTGTTGTCCGTTGCGCGTCTTGCGGTAACCGCATTACGGC
ATTCAAAGATGAAGCTGAACTCGAGtatagtgagtcgtattaattt

## Fragment 20 : *agmX*[1-90]

gccgaccgactgagacgctcacaagagctcTGATGCGCTTTGTCTGTGATAGCTGCCGCGCCCAGTATATGAT
CTCGGACGACAAGATCGGCCCGAAGGGCGTCAAGGTCCGTTGCAAGAAGTGCGGCCATACCAT
CACCGTCCGGCCCGCCGGCGCGACCGCGGCCAAGGATTCCGCGTCGGAAAGCAGCACCTCGG

AGGCCTCGGCGTCGACCGACGTCGGCAAGGGCTCGGATGCGTCCGCCGCGACGATGCCGGCG
ACCCTGGGCACCtgaattctatagtgagtcgtattaattt

## Acknowledgements

We thank Lucy Shapiro for materials and Antonio Frandi for help with ChIP-Seq analyses. Funding support is from the Swiss National Science Foundation (to PHV, to FHA and to SM) and the Novartis Consumer Health Foundation (to MB).

## Additional information

### Funding

| Funder | Grant reference number | Author |
| --- | --- | --- |
| Schweizerischer Nationalfonds zur Förderung der Wissenschaftlichen Forschung | 31003A_162716 | Patrick H Viollier |

The funders had no role in study design, data collection and interpretation, or the decision to submit the work for publication.

### Author contributions

MB, SC, JM, Conception and design, Acquisition of data, Analysis and interpretation of data, Drafting or revising the article, Contributed unpublished essential data or reagents; SH, Conception and design, Acquisition of data, Analysis and interpretation of data, Contributed unpublished essential data or reagents; LT, Acquisition of data, Contributed unpublished essential data or reagents; SM, Analysis and interpretation of data, Drafting or revising the article; FH-TA, PHV, Conception and design, Analysis and interpretation of data, Drafting or revising the article

### Author ORCIDs

Johann Mignolet, http://orcid.org/0000-0002-3721-4307
Seamus Holden, http://orcid.org/0000-0002-7169-907X
Patrick H Viollier, http://orcid.org/0000-0002-5249-9910

## Additional files

### Supplementary files

• Supplementary file 1. NMR and refinement statistics of ZitP$^{(1-43)}$. Data used to derive the NMR solution structure.

• Supplementary file 2. List of strains, plasmids and oligonucleotides. Details on strain and plasmid constructions and the oligonucleotides that were used.

### Major datasets

The following dataset was generated:

| Author(s) | Year | Dataset title | Dataset URL | Database, license, and accessibility information |
| --- | --- | --- | --- | --- |
| Matthieu Bergé, Patrick H Viollier | 2016 | ChIP-Seq | https://www.ncbi.nlm.nih.gov/geo/query/acc.cgi?acc=GSE79918 | Publicly available at the NCBI Gene Expression Omnibus (accession no. GSE79918) |

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
