## [Decision Letter]

Thank you for submitting your article "Modularity and determinants of a (bi-)polarization control system from free-living and obligate intracellular bacteria" for consideration by *eLife*. Your article has been reviewed by two peer reviewers, and the evaluation has been overseen by a Reviewing Editor and Vivek Malhotra as the Senior Editor. The following individual involved in review of your submission has agreed to reveal his identity: Grant Bowman (Reviewer #3).

The reviewers have discussed the reviews with one another and the Reviewing Editor has drafted this decision to help you prepare a revised submission

Summary:

The reviewers find that the identification and characterization of ZitP are highly interesting and contribute to our understanding of cell polarity in bacteria. The experiments are, in general, well-designed and the paper is clearly written. The very data rich paper demonstrates that ZitP is a factor that contributes to the positioning of PopZ. However, the conclusions on its role in origin segregation are less well supported and the authors make a few overinterpretations of their results, and these should be corrected before the manuscript is ready for publication. Some of these interpretations would be strengthened by additional supporting experiments, and others would be best left out of the manuscript.

Central conclusions:

1) The Zn-finger domain of ZitP is sufficient for interaction with PopZ, whereas its association with the membrane is critical for its function.

2) ZitP is found to associate, directly or indirectly, with sites flanking the PopZ/ParB-associated centromeric *parS* sites.

3) Deletion of ZitP or overexpression leads to defects in PopZ and ParB localization.

4) ZitP is able to mediate the redistribution of PopZ to both cell poles upon co-overexpression with PopZ in *C. crescentus* and in a heterologous *E. coli* system.

5) Similar properties are observed for ZitP homologues from various α-proteobacterial species, suggesting that the function of the protein is widely conserved in this lineage.

Essential revisions:

The reviewers raise a number of concerns that must be adequately addressed before the paper can be accepted. Some of the required revisions will likely require further experimentation.

In general we propose to remove some of the data concerning the claim that ZitP affects chromosome segregation as the current data does not fully supports this. The manuscript says important things about PopZ and cell polarity, including issues related to chromosome segregation. Focusing the paper on the ZitP-PopZ part will also make the paper more accessible to non Caulobacter readers.

Thus, instead of doing the proposed experiments listed below, some of the points listed can be solved by either downplaying the claims made and/or removing the data.

1) The conclusion that ZitP has a role in origin segregation that goes beyond controlling PopZ localization is not supported by the data:

A) The ChIP-seq results may be explained by the fact that ZipP is associated with the membrane-proximal surface of the PopZ complex and, therefore, located in the vicinity of the DNA regions flanking the PopZ-associated ParB-*parS* complex. The results obtained for PopZ may reflect a similar situation: crosslinking to the *parS* region itself may be less efficient because it requires the isolation of a tripartite complex consisting of PopZ, ParB and *parS* DNA, whereas crosslinking to the flanking regions that are not covered by ParB but still in the immediate proximity of PopZ may be more effective. The experiment using the mCherry-PopZ strain and an anti-RFP antibody (Figure 4, bottom row) should be performed in a strain lacking ZitP.

B) The synthetic effects caused by the deletion of zitP in the different popZ mutant backgrounds do not necessarily indicate a direct role of ZitP in ParAB regulation. All the defects in ParB localization observed are likely explained by the aberrant formation and localization of PopZ clusters, which in turn may affect the positioning of ParB-*parS* complexes or the localization/function of ParA and, thus, the control of cell division by the MipZ system. As shown by Ptacin et al. (2014), the mutant forms of PopZ are not fully defective in ParA/ParB binding but still show some affinity for the two proteins. Thus, the deletion of ZitP and the consequent changes in the efficiency/reliability of PopZ localization may simply aggravate the defects caused by the mutation of PopZ and thus enhance the segregation defects of the mutant strains. In agreement with this notion, the severity of the phenotype in all cases corresponds to the severity of the defect in PopZ localization (Figure 4—figure supplement 1). Similarly, the effects of zitP overexpression (stalling of ParB movement, displacement of monomeric ParA) may all be explained by the strongly aberrant PopZ localization patterns induced in this condition and their effects on ParB localization and ParA function. In my eyes, there is no evidence for the statement that "ZitP controls ParAB through a new mechanism that does not involved the known ParAB interaction sites in PopZ but unknown regions flanking the centromere".

2) Figure 2—figure supplement 1: The elution profile for ZitP needs to be shown to allow a definitive conclusion on the interaction between PopZ and ZitP.

3) The main function of ZitP appears to be the attachment of PopZ to the membrane, which in turn helps to establish the bipolar PopZ localization pattern. Is PopZ in the *C. crescentus* Δ*zitP* mutant no longer membrane-associated (even though its subcellular distribution is largely unchanged)? This could be clarified by PALM analysis of suitable *C. crescentus* strains. Similarly, it would be interesting to see that in *E. coli*, PopZ clusters are cytoplasmic and no longer membrane-associated (as suggested in the subsection “ZitP imparts bipolarity upon PopZ in *E. coli*”, first paragraph).

4) The results shown for RpZitP and RpPopZ in Figure 6 are not as clear as for those obtained for the *C. crescentus* proteins. The data obtained for the rickettsial proteins should be quantified.

5) From the evidence presented in this manuscript, it is clear that PopZ directs the localization of ZitP. However, the authors also strive to prove the reverse – that ZitP directs PopZ localization. They succeed in showing this under two circumstances – co-expression of proteins in *E. coli* and in the context of overproduction of ZitP^1-133^ in Caulobacter. At times, the authors seem to be drawing too strong of a conclusion from these results, as the same might be expected of any protein that binds to PopZ. Thus, ZitP's role in directing PopZ may not be unique, especially if PopZ has many binding partners. The possibility of many binding partners may be inferred from the fact that nearly all ST proteins (in addition to the chromosome centromere and ParA/B are delocalized in a *popZ* knockout background (Bowman et al. 2010). To support the claim that ZitP has a physiological role in directing PopZ localization, they also show that zitP knockouts enhance the cell division phenotype in PopZ mutant backgrounds where PopZ cannot interact with ParA/B. This is a critical result that bears further exploration. First, the authors should determine whether interaction with PopZ is needed for rescue by testing the effects of rescuing the PopZ^KE/KEP^ mutant with the W35I mutant of ZitP. Additionally, the authors should ask which parts of ZitP are critical for the rescuing effect by adding back different sections of ZitP into the PopZ^KE^ and/or KEP background. If ZitP^1-133^is sufficient to rescue the phenotype and ZitP^1-43^ is not, this suggests that membrane anchoring and not just interaction with PopZ is important for proper polar localization of PopZ. If full length ZitP is required, it suggests an important functional role for the C-terminal domain. If ZitP^1-43^ is sufficient, the model that ZitP directs PopZ localization will likely require some adjustment.

6) Chromatin-IP starts with formaldehyde x-linking, which binds any near neighbors to DNA. Thus, interaction between ZitP and DNA may not be direct or even indirect in the formal sense. It is likely that indirect binding to DNA was observed for PopZ, as sequence-dependent association with DNA probably occurs through ParB/ParA. However, we also know that centromeres reside near poles in Caulobacter cells, and that the poles also have PopZ. Thus, some of the binding observed by ChIP-seq may have occurred merely because PopZ and the centromere are in proximity – not from a direct or even an indirect connection between protein and DNA. Similarly, it seems possible that ZitP is merely in the vicinity of centromere by virtue of its interaction with PopZ, and not indirectly binding to DNA. The conclusions would be strengthened if the authors could provide more compelling evidence that ZitP isn't interacting with sites near centromeric DNA because it is at the poles and happens to be near DNA that is not otherwise occluded by ParA/ParB/PopZ. If this is not possible, several of the authors' conclusions should be reworded, the issue should be raised in the main text of the manuscript, and mechanisms outside of ZitP-DNA interactions should be discussed.

---

## [Author Response]

*Essential revisions:*

*The reviewers raise a number of concerns that must be adequately addressed before the paper can be accepted. Some of the required revisions will likely require further experimentation.*

In general we propose to remove some of the data concerning the claim that ZitP affects chromosome segregation as the current data does not fully supports this. The manuscript says important things about PopZ and cell polarity, including issues related to chromosome segregation. Focusing the paper on the ZitP-PopZ part will also make the paper more accessible to non Caulobacter readers.

*Thus, instead of doing the proposed experiments listed below, some of the points listed can be solved by either downplaying the claims made and/or removing the data.*

*1) The conclusion that ZitP has a role in origin segregation that goes beyond controlling PopZ localization is not supported by the data:*

A) The ChIP-seq results may be explained by the fact that ZipP is associated with the membrane-proximal surface of the PopZ complex and, therefore, located in the vicinity of the DNA regions flanking the PopZ-associated ParB-parS complex. The results obtained for PopZ may reflect a similar situation: crosslinking to the parS region itself may be less efficient because it requires the isolation of a tripartite complex consisting of PopZ, ParB and parS DNA, whereas crosslinking to the flanking regions that are not covered by ParB but still in the immediate proximity of PopZ may be more effective. The experiment using the mCherry-PopZ strain and an anti-RFP antibody (Figure 4, bottom row) should be performed in a strain lacking ZitP.

We agree and we now de-emphasize this aspect and just show that ZitP associates near *parS* and that this is dependent on PopZ and the N-terminal region that is required that is required for efficient binding of ParAB in *popZ^KE^*cells and in *popZ*∆^26^cells. While we still summarize these findings briefly, they are just used as supporting evidence and therefore placed it in Figure 4—figure supplement 1. We also determined that the association of PopZ with the *parS* region is not influenced by the Δ*zitP* mutation but chose to eliminate this from the manuscript.

*B) The synthetic effects caused by the deletion of zitP in the different popZ mutant backgrounds do not necessarily indicate a direct role of ZitP in ParAB regulation. All the defects in ParB localization observed are likely explained by the aberrant formation and localization of PopZ clusters, which in turn may affect the positioning of ParB-parS complexes or the localization/function of ParA and, thus, the control of cell division by the MipZ system. As shown by Ptacin et al. (2014), the mutant forms of PopZ are not fully defective in ParA/ParB binding but still show some affinity for the two proteins. Thus, the deletion of ZitP and the consequent changes in the efficiency/reliability of PopZ localization may simply aggravate the defects caused by the mutation of PopZ and thus enhance the segregation defects of the mutant strains. In agreement with this notion, the severity of the phenotype in all cases corresponds to the severity of the defect in PopZ localization (Figure 4—figure supplement 1). Similarly, the effects of zitP overexpression (stalling of ParB movement, displacement of monomeric ParA) may all be explained by the strongly aberrant PopZ localization patterns induced in this condition and their effects on ParB localization and ParA function. In my eyes, there is no evidence for the statement that "ZitP controls ParAB through a new mechanism that does not involved the known ParAB interaction sites in PopZ but unknown regions flanking the centromere".*

We agree with this comment that the effect of ZitP on DNA segregation are mediated by its effect on PopZ localization and we modified the corresponding passage, concluding that ZitP is able to redirect PopZ localization via direct interaction between the two partners and ZitP membrane anchoring.

*2) Figure 2—figure supplement 1: The elution profile for ZitP needs to be shown to allow a definitive conclusion on the interaction between PopZ and ZitP.*

Agreed. This is fixed now and the omitted elution profile added to Figure 2—figure supplement 1 unambiguously that the two proteins co-elute together, indicative of a direct interaction that we further quantify by ITC in Figure 2.

*3) The main function of ZitP appears to be the attachment of PopZ to the membrane, which in turn helps to establish the bipolar PopZ localization pattern. Is PopZ in the C. crescentus ΔzitP mutant no longer membrane-associated (even though its subcellular distribution is largely unchanged)? This could be clarified by PALM analysis of suitable C. crescentus strains. Similarly, it would be interesting to see that in E. coli, PopZ clusters are cytoplasmic and no longer membrane-associated (as suggested in the subsection “ZitP imparts bipolarity upon PopZ in E. coli”, first paragraph).*

To address this question, we performed some fractionation experiment in *E. coli* as well as in *C. crescentus* to see if the presence of ZitP could influence the fractionation properties of PopZ in extracts. To do so, we separated extracts from the relevant under- and over-expression strains in soluble and insoluble fractions by centrifugation and then treated the insoluble fraction with different detergents to test if PopZ can be solubilized. Unfortunately, in the presence or absence of ZitP in *C. crescentus* as well as in *E. coli*, the PopZ fractionation profile did not reveal any difference as in all cases PopZ accumulates in the soluble fraction to the same extent. Perhaps PopZ interactions with its partners are labile and disrupted during extract preparation. As the blots were not revealing we did not include them in the manuscript, but we include the data for the fractionation done with proteins expressed in *E. coli* as Figure 7 (labels are as for Figure 5—figure supplement 1 where fractionation of ZitP is shown).

Author response image 1.**DOI:**
http://dx.doi.org/10.7554/eLife.20640.018

The suggested two-color PALM experiments in live cells that would be necessary to reveal the difference between polar membrane-localized PopZ versus polar membrane proximal PopZ could not be done for technical reasons. Additionally, we decided not to push hard to find a solution because it seemed unlikely that a difference would be visible even if the technical problems could be resolved. While it is certainly plausible that ZitP acts as membrane anchor for PopZ, it clearly does act as sole membrane anchor since there are other transmembrane proteins such as DivL and CckA as reported by the Bowman group that interact with PopZ directly and would likely mask the contribution (i.e. membrane association) by ZitP.

*4) The results shown for RpZitP and RpPopZ in Figure 6 are not as clear as for those obtained for the C. crescentus proteins. The data obtained for the rickettsial proteins should be quantified.*

We now show that membrane-anchored Rickettsial ZitP can induce the formation of bipolar plugs of *C. crescentus* PopZ in *C. crescentus* (Figure 6) and in *E. coli* (Figure 6). Moreover, Dendra-tagged *C. crescentus* ZitP (full length) can also adopt a bipolar disposition when co-expressed with rickettsial PopZ in *E. coli* (Figure 6, this is the same figure as shown before, whereas 6B and 6E are new panels).

*5) From the evidence presented in this manuscript, it is clear that PopZ directs the localization of ZitP. However, the authors also strive to prove the reverse – that ZitP directs PopZ localization. They succeed in showing this under two circumstances – co-expression of proteins in E. coli and in the context of overproduction of ZitP^1-133^ in Caulobacter. At times, the authors seem to be drawing too strong of a conclusion from these results, as the same might be expected of any protein that binds to PopZ. Thus, ZitP's role in directing PopZ may not be unique, especially if PopZ has many binding partners. The possibility of many binding partners may be inferred from the fact that nearly all ST proteins (in addition to the chromosome centromere and ParA/B are delocalized in a popZ knockout background (Bowman et al. 2010). To support the claim that ZitP has a physiological role in directing PopZ localization, they also show that zitP knockouts enhance the cell division phenotype in PopZ mutant backgrounds where PopZ cannot interact with ParA/B. This is a critical result that bears further exploration. First, the authors should determine whether interaction with PopZ is needed for rescue by testing the effects of rescuing the PopZ KE / KEP mutant with the W35I mutant of ZitP. Additionally, the authors should ask which parts of ZitP are critical for the rescuing effect by adding back different sections of ZitP into the PopZ KE and/or KEP background. If ZitP*^*1-133*^
*is sufficient to rescue the phenotype and ZitP*^*1-43*^
*is not, this suggests that membrane anchoring and not just interaction with PopZ is important for proper polar localization of PopZ. If full length ZitP is required, it suggests an important functional role for the C-terminal domain. If ZitP*^*1-43*^
*is sufficient, the model that ZitP directs PopZ localization will likely require some adjustment.*

The proposed experiment was done and the results (reported as growth curves) are shown in Figure 4. The figure shows complementation experiment with the different variants as Dendra-ZitP fusions expressed from the *xylX* locus (ZitP^1-43^, ZitP^1-133^ and ZitP full-length) in the *popZ^KE^*background. Only ZitP full-length and the ZitP anchored to the membrane (ZitP^1-33^) are able to promote growth of the mutant cells. By contrast, when ZitP can no longer interact with PopZ (i.e. the ZitP^CS^ variant) or when only soluble (cytoplasmic) part of ZitP is expressed (ZitP^1-43^), cells grow poorly.

These results not only demonstrate that the direct interaction of ZitP with PopZ is required but also that the membrane anchoring is essential for ZitP to control PopZ.

6) Chromatin-IP starts with formaldehyde x-linking, which binds any near neighbors to DNA. Thus, interaction between ZitP and DNA may not be direct or even indirect in the formal sense. It is likely that indirect binding to DNA was observed for PopZ, as sequence-dependent association with DNA probably occurs through ParB/ParA. However, we also know that centromeres reside near poles in Caulobacter cells, and that the poles also have PopZ. Thus, some of the binding observed by ChIP-seq may have occurred merely because PopZ and the centromere are in proximity – not from a direct or even an indirect connection between protein and DNA. Similarly, it seems possible that ZitP is merely in the vicinity of centromere by virtue of its interaction with PopZ, and not indirectly binding to DNA. The conclusions would be strengthened if the authors could provide more compelling evidence that ZitP isn't interacting with sites near centromeric DNA because it is at the poles and happens to be near DNA that is not otherwise occluded by ParA/ParB/PopZ. If this is not possible, several of the authors' conclusions should be reworded, the issue should be raised in the main text of the manuscript, and mechanisms outside of ZitP-DNA interactions should be discussed.

With the new ChIP-seq data it is now clear that PopZ, but not ZitP, is required for binding of the PopZ complex at these two (*parS*-flanking) sites. Moreover, the interaction of PopZ with ParA/B (or other DNA binding proteins that might associate with PopZ N-terminal region) is clearly critical as well. However, because the binding is distinct from that of ParB it is possible that this mediated via ParAB binding to adjacent sites or by another (unknown) protein in the PopZ complex. The ChIP-seq results are de-emphasized in the modified Results section, but briefly mentioned in the revised Discussion.